

# Anthropogenic and internal drivers of wind changes over the Amundsen Sea, West Antarctica, during the 20th and 21st centuries

Paul R. Holland[1], Gemma K. O'Connor[2], Thomas J. Bracegirdle[1], Pierre Dutrieux[1], Kaitlin A. Naughten[1], Eric J. Steig[2], David P. Schneider[3,4], Adrian Jenkins[5], and James A. Smith[1]

[1]British Antarctic Survey, High Cross, Madingley Road, Cambridge, CB3 0ET, UK.
[2]University of Washington, Seattle, USA.
[3]National Center for Atmospheric Research, Boulder, USA.
[4]University of Colorado, Boulder, USA.
[5]Northumbria University, Newcastle, UK.

*Correspondence to*: Paul Holland (p.holland@bas.ac.uk)

**Abstract.** Ocean-driven ice loss from the West Antarctic Ice Sheet is a significant contributor to sea-level rise.
Recent ocean variability in the Amundsen Sea sector is primarily controlled by near-surface winds. We combine paleoclimate reconstructions and climate model simulations to understand past and future influences on Amundsen Sea winds from anthropogenic forcing and internal climate variability. The reconstructions show strong historical wind trends. External forcing from greenhouse gases and stratospheric ozone depletion drove zonally-uniform westerly wind trends centred over the deep Southern Ocean. Internally-generated trends resemble
a south Pacific Rossby wave train, and were highly influential over the Amundsen Sea continental shelf. There was strong interannual and interdecadal variability over the Amundsen Sea, with periods of anticyclonic wind anomalies in the 1940s and 1990s, when rapid ice loss was initiated. Similar anticyclonic anomalies probably occurred prior to the 20th century, but without causing the present ice loss. This suggests that ice loss may have been triggered naturally in the 1940s, but failed to recover subsequently due to the increasing importance of
anthropogenic forcing from greenhouse gases (since the 1960s) and ozone depletion (since the 1980s). Future projections also feature strong wind trends. Emissions mitigation influences wind trends over the deep Southern Ocean but has less influence on winds over the Amundsen Sea shelf, where internal variability creates a large and irreducible uncertainty. This suggests that strong emissions mitigation is needed to minimise ice loss this century, but that the uncontrollable future influence of internal climate variability could be equally important.

**1 Introduction**

The West Antarctic Ice Sheet (WAIS) is losing ice at an increasing rate (Shepherd et al., 2019) and forms a major source of uncertainty in projections of global sea-level rise (IPCC, 2019). The most rapid ice loss is occurring in the Amundsen Sea sector, where thinning and retreat of the floating ice shelves is causing acceleration of their tributary glaciers (De Rydt et al., 2021). The ice loss is caused by changes in melting of ice shelves by the ocean
(Shepherd et al., 2004), but it is not clear why this melting has increased, leading to the current ice-sheet mass imbalance.

Several lines of evidence document the history of ice loss in this region. A synthesis of geological and geophysical datasets implies that the ice sheet geometry was broadly stable in this region for ~10,000 years prior to the current





retreat (Larter et al., 2014). Sediment records from beneath Pine Island Glacier ice shelf show that its grounding line started to retreat from a prominent seabed ridge in the 1940s, with the ice shelf finally detaching from this ridge in the 1970s (Jenkins et al., 2010; Smith et al., 2017). Ice velocity records from satellite data suggest that Amundsen Sea glacier discharge was accelerating from the earliest observations in the 1970s (Mouginot et al., 2014), while satellite altimetry and interferometry data show glacier thinning and grounding-line retreat occurring

since at least the early 1990s (Rignot et al., 2014; Paolo et al., 2015; Konrad et al., 2017). Glacier discharge and thinning have accelerated markedly in recent decades (Mouginot et al., 2014; Paolo et al., 2015; Konrad et al., 2017; Shepherd et al., 2019). Overall, this evidence suggests that ice loss was triggered in the 1940s and evolved slowly but continually thereafter, until accelerating from the 1990s onwards.

There are several possible explanations for this history of ice loss. First, there is evidence that the 1940s ice retreat was triggered by a period of strong climate variability associated with the tropical Pacific (Schneider and Steig, 2008; Steig et al., 2012; Steig et al., 2013; Jenkins et al., 2018; Holland et al., 2019). If this is the sole cause of the changes the climatic anomaly must have been an extremely unusual event, because the ice sheet was previously stable for millennia in this region (Larter et al., 2014). Second, it is likely that ongoing ice loss is

sustained by a range of ice and ocean feedbacks (De Rydt et al., 2014; Favier et al., 2014; Lhermitte et al., 2020; Bett et al., 2020). However, the WAIS cannot be experiencing a purely unstable, self-sustaining retreat because the rate of ice discharge remains responsive to ocean variability (Christianson et al., 2016; Jenkins et al., 2018). Finally, an overall warming of the Amundsen Sea over the 20th century may have caused an increase in melting that sustains the current ice loss (Holland et al., 2019; Naughten et al., 2022). Any such centennial change could

be caused by a combination of anthropogenic forcing and internal climate variability. All of these different processes may be contributing to the ongoing ice loss, but the relative contributions of a historical trigger, ice-ocean feedbacks, and changes in climatic forcing are not yet known.

The Amundsen Sea is subject to strong wind-driven variability – primarily linked to the tropical Pacific (Lachlan-
Cope and Connolley, 2006; Ding et al., 2011; Steig et al., 2012) – that is clearly reflected in ice-shelf melting (Dutrieux et al., 2014; Jenkins et al., 2018) and the ice-sheet response (Christianson et al., 2016; Jenkins et al., 2018). Variable winds modulate the supply of relatively warm Circumpolar Deep Water onto the continental shelf, driving decadal anomalies in ocean thermocline depth and ice-shelf melting (Thoma et al., 2008; Kimura et al., 2017). This strong decadal variability means that any trends caused by anthropogenic climate forcing may not be

detectable in the short ocean and ice sheet observational records (commencing in the 1990s) or atmospheric reanalysis data (reliable only since 1979).

Climate model simulations suggest that a gradual eastward wind trend occurred over the Amundsen Sea shelf break during the 20$^{th}$ century (Holland et al., 2019). Such westerly wind trends are a very well-established response

of the Southern Hemisphere climate to anthropogenic forcings (Arblaster and Meehl, 2006; Son et al., 2010; Thompson et al., 2011; Gillett et al., 2013; Bracegirdle et al., 2014; Bracegirdle et al., 2020; Goyal et al., 2021). In ocean simulations, this wind trend drives an increased prevalence of warm decadal ocean anomalies and an increase in ice shelf melting (Naughten et al., 2022), providing the first evidence that anthropogenic forcings may have contributed to ice loss from the WAIS. However, the importance of these externally-forced westerly wind



trends has recently been challenged by paleoclimate reconstructions (Dalaiden et al., 2021; O'Connor et al., 2021). These reconstructions show that a deepening of the Amundsen Sea Low dominates the local wind trends, leading to westward trends over the Amundsen Sea shelf. The larger-scale westerly trends are also found in the reconstructions, but shifted further north. An analogous deepening of the Amundsen Sea Low in recent decades (since 1979) is thought to be primarily driven by natural tropical Pacific variability (Raphael et al., 2016; Meehl

et al., 2016; Purich et al., 2016; Schneider and Deser, 2018). Therefore, the reconstructions suggest that 20th century wind trends over the Amundsen Sea shelf, associated with the Amundsen Sea Low deepening pattern, were primarily internally generated. Thus, the contribution of anthropogenic forcings to centennial wind changes in this region remains unclear.

It is crucial to quantify the contribution of anthropogenically-forced climate change to past and future ice loss from the WAIS. If the ice loss is dominated by internal climate variability and ice/ocean feedbacks, it may be unavoidable and/or irreversible on centennial timescales. If the ice loss has a substantial anthropogenic component, it may respond to future reductions in anthropogenic forcing. Quantifying the interplay between these factors is therefore crucial to adaptation and mitigation decisions. It is also important to distinguish the influence

of different anthropogenic forcings (e.g. greenhouse gases (GHGs) versus ozone depletion), as their mitigation is different. In this study, we combine paleoclimate reconstructions and climate model simulations to investigate historical wind trends over the Amundsen Sea and their attribution to different forcings, and projected future wind trends and their sources of uncertainty. Since wind-forced ocean variability is known to influence ice loss in the Amundsen Sea, understanding these wind changes is informative about the drivers of change in the WAIS.

## 2 Methods

### 2.1 Paleoclimate reconstructions

Paleoclimate reconstructions are used to understand historical changes in wind forcing over the Amundsen Sea, which arise through the combined influence of external forcing and internal variability. We use 1°-resolution annually-resolved reconstructions of near-surface zonal winds and sea-level pressure (SLP) spanning 1900 to

2005 from O'Connor et al. (2021). In addition, for this study, we extend the technique to reconstruct near-surface meridional winds and sea-surface temperatures (SSTs). The reconstructions are created using an offline data assimilation method that combines the temporal variability from a sparse set of climate proxies with the spatial covariance fields obtained from climate model simulations. Annually-averaged anomaly states (relative to the 1961-1990 mean) are randomly drawn from climate-model output to form a 'prior' ensemble, and this prior is

used for every year reconstructed (Hakim et al., 2016; Tardif et al., 2019). This ensures that all temporal variance and trends in the final reconstruction are generated from the proxy data rather than the climate model. Proxy data used here include the global PAGES2k proxy database (PAGES2k, 2017), supplemented with additional ice-core accumulation and coral records. This means that the reconstructions are well-constrained over West Antarctica and the tropical Pacific, the crucial regions of importance to this study.


O'Connor et al. (2021) assess the interannual variability in their reconstructed SLP and zonal winds against reanalysis fields. Owing to the availability of proxy data used in the reconstructions and the dominant climate



modes in this region, the reconstructions are most skilful over the south Pacific. The reconstructions are also qualitatively in agreement with those of Dalaiden et al. (2021), who use a different data assimilation method and a proxy database focussed on the Southern high latitudes. O'Connor et al. (2021) assess the different sources of uncertainty in the reconstructions and pay particular attention to the influence of the climate simulations used as the prior, finding that reconstructed trends are sensitive to the inclusion of anthropogenic forcing in the prior simulations. In this study, we focus on reconstructions using the Community Earth System Model (CESM1) 'Pacific Pacemaker' simulations as the prior. This is an ensemble of historical simulations that contain all anthropogenic forcings but also have eastern tropical Pacific SSTs constrained to follow observations – the 'pacemaker' (Schneider and Deser, 2018). We favour these simulations because they should best represent both externally-forced and Pacific modes of variability. O'Connor et al. (2021) show that this prior creates reconstructions with high skill throughout the south Pacific, and we find that the reconstructions perform well when compared to ERA5 reanalysis wind anomalies for the timeseries of interest in this study (see section 3.1.3 below).

We found that reconstructed near-surface wind trends near the coastal regions of the Amundsen Sea are noisy, and deviate significantly from geostrophic winds derived from the reconstructed SLP. The derived geostrophic wind anomalies are also better correlated with ERA5 reanalysis surface wind anomalies. This is unsurprising because the reconstructed SLP patterns show greater skill than near-surface winds (O'Connor et al., 2021) and should better reflect the large-scale climate modes that are constrained by the remote proxies. Near-surface winds are driven by the same SLP of course, but are also influenced by uncertain features of the reconstructed near-surface atmospheric structure. Therefore, throughout this study we use geostrophic wind anomalies from the reconstructed SLP. To make these geostrophic winds most comparable to the climate model winds considered next, a simple near-surface correction was derived by comparing geostrophic and near-surface winds in the CESM1 Large Ensemble, which is described below. The correction consists of rotating the geostrophic winds by 10 degrees clockwise and multiplying their amplitude by 0.9. This correction marginally improved the correlation of annual wind anomalies to ERA5 reanalysis winds (see section 3.1.3).

### 2.2 Climate model simulations

Climate simulations are used to understand the role of anthropogenic forcing in the past and future. By comparing these simulations with the reconstructions, we are also able to estimate the historical role of internal climate variability. We consider near-surface winds, SLP and SST from a total of 180 simulations within ten ensembles of simulations using CESM1 under different forcings (Kay et al., 2015; England et al., 2016; Sanderson et al., 2017; Sanderson et al., 2018; Schneider and Deser, 2018; Deser et al., 2020). These simulations use the CAM5 atmospheric model, POP2 ocean model, and CICE4 sea ice model, all run at approximately 1° resolution (Hurrell et al., 2013; Kay et al., 2015). The ensembles are fully described in Table 1. Near-surface wind components were not available on standard height levels, so we used winds at the bottom pressure level of the atmospheric model, whose height varies in time and space, but is around 50 m over our domain. The atmospheric model uses a 0.9° latitude × 1.2° longitude grid, so all winds are binned onto a polar stereographic grid with 200 km resolution to clarify the plotting over the region of interest. SLP and SST fields are considered on the native model atmosphere and ocean grids, respectively.



Many features of this study are only possible because such a wide range of simulations are available for CESM1. The use of a single model is also necessary to ensure that the responses studied are driven only by the prescribed

forcings. However, this also means that our conclusions do not account for model structural uncertainty. Fortunately, the CESM1 features a good representation of winds over the Amundsen Sea (Holland et al., 2019) and contains a faithful representation of the Amundsen Sea Low (England et al., 2016) and its crucial teleconnection to the tropical Pacific (Holland et al., 2019). It also produces a good general representation of a wide range of other variables around Antarctica, including sea-surface temperatures, sea ice, sea-level pressure,

ice-sheet surface mass balance (Agosta et al., 2015; Lenaerts et al., 2016), and ocean conditions in the Amundsen Sea (Barthel et al., 2020). However, CESM1 has some biases that are pertinent to the present study. The westerly winds and absorbed shortwave radiation over the Southern Ocean are both too strong (Kay et al., 2016), and this may bias the model response to external forcing, as suggested by the inability of the model to reproduce recent sea ice trends (Schneider and Deser, 2018). These biases should be kept in mind as a source of uncertainty in

using the model results to separate the forced and internal components of this circulation history. Crucially, however, the use of CESM1 simulations as the prior model for the reconstruction ensures that our methodology is internally consistent in this regard.

This study considers trends during the period 1920-2080, for which most simulations are available (Table 1). This

is divided into two equal 'historical' (1920-2000) and 'future' (2000-2080) periods, which are appropriate to climatic forcing of the WAIS. The year 1920 pre-dates the recent ice-sheet retreat (Smith et al., 2017), a modelled warming of the Amundsen Sea (Naughten et al., 2022), and most anthropogenic climate forcing (Kay et al., 2015). The WAIS was rapidly losing mass well before 2000 (Mouginot et al., 2014; Konrad et al., 2017). Therefore, any changes during 1920-2000 are pertinent to the attribution of the ongoing ice loss. The full consequences of ice-

sheet melting will take centuries to emerge, but climatic forcing in the decades prior to 2080 are the subject of immediate adaptation and mitigation decisions.

The central ensemble considered is the CESM1 'Large Ensemble' (LENS) (Kay et al., 2015), a perturbed-initial-condition ensemble from 1920-2100 using known external forcings (GHGs, stratospheric ozone depletion,

aerosols, land-use change, solar, volcanic) up to 2005 and the strong radiative forcing RCP8.5 scenario thereafter. All other ensembles use the same model as LENS, but with different forcings. By calculating the differences between ensembles with and without particular forcings, we are able to isolate the local climate 'responses' to those forcings.

The 'Pacific Pacemaker' ensemble (PACE) is used in the reconstruction prior. PACE uses the same external forcings as LENS, but is additionally constrained to follow the observed history of SST anomalies in the central and eastern tropical Pacific Ocean (Schneider and Deser, 2018). In a previous study (Holland et al., 2019), PACE was used to estimate the role of internal variability associated with the tropical Pacific. In the present study this function is performed by the paleoclimate reconstructions, which allow us to estimate the influence of variability

from all sources, Pacific and otherwise. This change in approach has important consequences for the results, as discussed below.





A 'pre-industrial control' ensemble (PIctrl) is used to assess the significance of externally-forced signals. PIctrl is constructed from a long simulation without external forcings (Kay et al., 2015), by splitting it into 22 non-overlapping 80-year periods, and then using the resulting ensemble for both the historical and future periods.

We consider four ensembles of projections under different future forcing scenarios. LENS experiences an extreme high forcing scenario, RCP8.5 (van Vuuren et al., 2011). The CESM1 'Medium Ensemble' (MENS) (Sanderson et al., 2018) follows RCP4.5, an intermediate forcing scenario. The CESM1 'low warming' ensembles (1.5degC and 2.0degC) are subjected to forcings designed never to exceed global warming of 1.5°C and 2°C above pre-industrial levels, targets that are associated with the Paris Agreement (Sanderson et al., 2017). In the CESM1, both scenarios require negative net carbon emissions before 2100. Each member of these future ensembles starts in 2005 and is a continuation of a historical member from LENS, so we concatenate the appropriate historical and future simulations in order to study 'future' trends from 2000.

Four further ensembles allow us to determine the individual influences of GHGs, ozone depletion, industrial aerosols, and biomass-burning aerosols (England et al., 2016; Deser et al., 2020). For each of these forcings, we consider an ensemble of 'leave-one-out' simulations in which the forcing is held fixed, while all other forcings evolve. Subtracting each of these ensembles from LENS isolates the response to each excluded forcing (Deser et al., 2020). For example, we introduce a 'no greenhouse gas' (XGHG) ensemble, which uses the same forcings as LENS apart from GHG concentrations, which are fixed at the 1920 level. We then derive the 'GHG response' by subtracting XGHG from LENS, yielding the influence of GHGs that are increasing in LENS but fixed in XGHG. This procedure is repeated for each of the forcings. An advantage of this 'leave-one-out' methodology is that the derived difference includes nonlinear interactions between the excluded forcing and all other forcings.

We consider a set of such climate 'responses', defined as the difference in ensemble-mean winds between two ensembles that have a difference in forcing. This has the advantage that the significance of a 'response' can be assessed by comparing the distributions of wind trends in the two ensembles. Specifically, we conduct a two-sample $t$-test with unequal sample size and variance to test the null hypothesis that the ensembles have the same mean trend, and treat the response as significant if that hypothesis is rejected at the 95% confidence level (two-sided). For example, the GHG wind trend response is significant if the distribution of LENS ensemble trends differs from the distribution of XGHG ensemble trends. We note that the number of simulations in each ensemble varies (Table 1) and this affects the significance of the derived responses.

## 3 Results

### 3.1 Historical changes

#### 3.1.1 Global patterns of change

Figure 1a shows a global map of the reconstructed historical trends in SST and SLP. We re-iterate that this reconstruction is most skilful over the south Pacific, the focus of this study, and may be less trustworthy in other regions (O'Connor et al., 2021). The reconstruction shows a widespread SST increase, particularly strong in the



eastern and central tropical Pacific, and a strengthening and southward shift of the Southern Hemisphere westerly winds (O'Connor et al., 2021). Crucially, there is a substantial regional deepening of SLP over the Amundsen Sea.

We interpret these features by separating their climatic drivers using the climate model simulations. Figure 1b shows the total externally-forced changes (from GHGs, ozone depletion, aerosols, land-use change, solar, and

volcanic sources) during this period, which is estimated as the ensemble mean of the LENS trends. The LENS members contain 40 different random realisations of internal climate variability, but they all have the same external forcing. Thus, if we average over all members the internal variability cancels out, and the externally-forced trends appear. This estimate agrees with many previous studies that historical external forcing drove broad background warming over the South Pacific (e.g. Deser et al., 2020; IPCC, 2019), and a zonally uniform

southward shift and acceleration of the westerlies (e.g. Arblaster and Meehl, 2006; Goyal et al., 2021).

We isolate the contribution of individual external forcings using the set of 'leave-one-out' ensembles, as described above. The 'GHG response' in Figure 1d is derived by subtracting the XGHG ensemble mean trends from the LENS ensemble mean trends, since the only difference between these ensembles is the increasing GHGs in LENS.

The historical increase in atmospheric GHG concentrations causes a strong SST warming (Figure 1d), which exceeds the total externally-forced warming (Figure 1b) because aerosol forcing drove cooling during this period (not shown), particularly in the Northern Hemisphere (Deser et al., 2020). GHG forcing drove over half of the historical changes in the westerly winds (Arblaster and Meehl, 2006; Gillett et al., 2013). Similarly deriving an 'ozone response' in Figure 1f by subtracting XOZO from LENS, we see that ozone depletion drove no strong SST

trends, but made a substantial contribution to the wind trends (Thompson et al., 2011; Son et al., 2010). The response to ozone depletion is slightly weaker than the response to GHGs because our chosen historical trend period includes many decades before the onset of rapid ozone depletion. Aerosols drove no substantial Southern Hemisphere wind trends over this period and are not considered further.

By subtracting the externally-forced changes in Figure 1b from the total reconstructed trends in Figure 1a, we may estimate the part of the trends caused by internally-generated variability (Figure 1c). This estimate reveals a coherent pattern of alternating SLP anomalies over the south Pacific that feature a prominent trend towards low pressure over the Amundsen Sea. These SLP trends are also supported by SST trends, with warm anomalies wherever the associated wind trend is southward, and cold anomalies wherever it is northward (Ciasto and

Thompson, 2008). This demonstrates that the reconstructed historical deepening of the Amundsen Sea Low (Dalaiden et al., 2021; O'Connor et al., 2021) is internally generated. This pattern of alternating SLP anomalies is suggestive of a Rossby wave train, a well-established mechanism for the poleward propagation of atmospheric anomalies over the south Pacific (Karoly, 1989; Lachlan-Cope and Connolley, 2006; Ding et al., 2011; Steig et al., 2012; Holland et al., 2019). However, the anomaly pattern in Figure 1c also contains features that are unusual.


It is well known that Pacific modes of variability – e.g. the El Niño—Southern Oscillation (ENSO) and Interdecadal Pacific Oscillation (IPO) – induce a global-scale atmospheric response that is highly influential over the Amundsen Sea (Karoly, 1989; Lachlan-Cope and Connolley, 2006; Ding et al., 2011; Steig et al., 2012; Dutrieux et al., 2014; Jenkins et al., 2018; Holland et al., 2019). Figure 1e illustrates the spatial structure of the



IPO response. This figure is constructed from the regression of monthly ERA5 reanalysis SLP and SST fields
(Hersbach, 2020) onto the IPO tripole index (Henley et al., 2015) during the period 1979-2020. This tripole index
is based on the temperature difference between tropical and subtropical Pacific SSTs (Henley et al., 2015) from
the HadISST1.1 dataset (Rayner et al., 2003) and has units of °C. To compare this panel with the others, it is
plotted as the response of SLP and SST to a hypothetical trend of -1 °C/century in the IPO index. (For example,

the hypothetical SLP trend (hPa/century) is calculated by multiplying the SLP regression onto the IPO index
(hPa/°C) by the hypothetical trend in the IPO index (-1 °C/century).) The illustrative value of -1 °C/century was
arbitrarily chosen to produce a deepening pressure over the Amundsen Sea (Figure 1e) that is comparable to the
internally-generated trend in the reconstruction (Figure 1c).

Comparing Figures 1c and 1e suggests that the reconstructed internally-generated trends cannot be simply related
to the classical IPO and ENSO modes of Pacific variability. Trends associated with these modes typically feature
a zonal dipole of SST and SLP anomalies over the tropical Pacific, and a Rossby wave train radiating southwards
from the western tropical Pacific (Figure 1e). Instead, the internally-generated trends appear to originate in the
sub-tropical south Pacific at approximately 30 °S (Figure 1c). Crucially, the deepening pressure trend over the

Amundsen Sea in the reconstruction occurs alongside a warming of the eastern tropical Pacific (Figure 1c), not
the cooling expected from IPO or ENSO variability (Figure 1e). We conclude that the internally-generated trends
are not associated with these classical modes, but do follow a similar (Rossby wave) propagation. Similar results
are obtained with an alternative paleoclimate reconstruction (Q. Dalaiden, personal communication, 2022). It is
not necessarily surprising that this centennial variability has a different pattern to interannual (ENSO) and

interdecadal (IPO) variability. Little is known about variability on these centennial timescales. It is important to
note that this discussion refers only to 80-year trends, which arise through the residual of many shorter IPO/ENSO
anomalies of opposing sign. This aspect is revisited in section 3.1.3 below.

**3.1.2 Winds over the Amundsen Sea**

Figure 2 has the same six panels as Figure 1, but with the analysis repeated for near-surface wind trends rather

than SLP and SST, and focussing on the Pacific sector of the Southern Ocean. The magenta boxes show three
areas of the Amundsen Sea that illustrate wind changes of potential relevance to the WAIS - shelf sea, shelf break,
and deep ocean. The wind trends reflect the SLP trends in Figure 1. Figure 2a shows the reconstructed historical
wind trends, with vectors coloured black if either the zonal or meridional wind trend is significant relative to the
interannual variability in the reconstruction. The reconstruction features significant westerly wind trends over

almost the entire Southern Ocean, apart from the Amundsen Sea shelf. A cyclonic pattern of wind trends is centred
on the Amundsen Sea shelf break, arising from the negative pressure trend in Figure 1a. This means that the
westerly trends to the north transition to easterly trends near the Amundsen Sea coast.

Figures 2b, 2d, and 2f show the externally-forced wind trends derived from the climate model simulations. These

figures are derived in the same way as their counterparts in Figure 1. Wind trend vectors are coloured black if
either the zonal or meridional wind trend is significant according to a two-sample $t$-test that compares the
distribution of trends within two ensembles. For example, in Figure 2b the externally-forced wind trends are
estimated as the LENS ensemble-mean trends. At each location, these are considered significant if the distribution





of trends in the LENS ensemble has a significantly different mean from the distribution of trends in the PIctrl
ensemble under the *t*-test. Figure 2d shows the GHG-induced trends derived by subtracting XGHG ensemble-
mean trends from LENS ensemble-mean trends, with vectors shown as significant if the distributions of trends in
LENS and XGHG have a different mean under the *t*-test. Figure 2f is the same for ozone.

Figures 2b, 2d, and 2f show that the historical westerly wind trends are clearly associated with radiative forcing
from GHGs and ozone depletion. (Arblaster and Meehl, 2006; Son et al., 2010; Thompson et al., 2011; Gillett et
al., 2013). In a feature that may be important to climatic forcing of the WAIS, these anthropogenic changes are
strongest over the deep ocean, but not significant over the Amundsen Sea shelf. The shelf break sits in a transition
zone, where externally-forced trends are significant overall (Figure 2b), but neither GHG nor ozone responses are
significant individually (Figures 2d and 2f). This spatial pattern of response to external forcings is also found in
the wider ensemble of climate models (Bracegirdle et al., 2014). The broader westerly trends are driven by
meridional gradients in atmospheric warming (Harvey et al., 2014; Shaw, 2019), but local controls on wind trends
over the Amundsen Sea are complex and require further study.

Over the time period considered, internal variability induces wind trends of comparable magnitude to the
externally-forced trends (Figure 2c). As expected from the deepening SLP trends described above, these
internally-generated wind trends have a cyclonic orientation, centred over the deep ocean north of the Amundsen
Sea. As a result, internal variability is associated with an easterly trend over the Amundsen Sea, in direct
opposition to the externally-forced westerly trend. The internally-generated trends are strongest on the shelf and
weaken to the north, in contrast to the externally-forced trends. Considering this region alone, the internally-
generated trends closely resemble those expected from the hypothetical negative trend in the IPO index (Figure
2e).

It is important to understand the level of consistency between the paleoclimate reconstruction and climate model
simulations, since the role of internal variability is derived by taking their difference. Since the reconstructions
use CESM1 simulations as their prior ensemble, in principle there is no structural difference between the two.
This can be illustrated by comparing the reconstruction to individual simulations from the LENS ensemble. The
LENS ensemble members all have the same externally-forced trends, but have 40 different realisations of the
trends generated by internal variability. Figure 3 shows the reconstruction (top row) alongside two ensemble
members chosen manually to illustrate the range of internal variability (other rows). For each of these sources,
both the total historical trends are shown (left panels) and the internally-generated part only (right panels). The
internally-generated trends are calculated, as before, by subtracting the externally-forced trends (the LENS
ensemble mean) from the total trends. The LENS ensemble members produce a wide variety of internally-
generated trends. By chance, LENS member 9 has a realisation of internal variability that produces cyclonic
trends, with a similar influence over the Amundsen Sea to the reconstruction. This is not unusual within the
ensemble, and illustrates the consistency between the reconstruction and LENS. By contrast, LENS member 18
exhibits anticylonic internally-generated trends. This highlights the importance of the reconstruction to this study.
Without the reconstruction we would have to treat all LENS members as being equally plausible estimates of the
real historical trends, and would not be able to constrain the important role of internal climate variability.



### 3.1.3 Temporal evolution of the anomalies

So far, we have only considered 80-year historical wind trends, but there is substantial variability in the winds on shorter timescales, and this is known to influence WAIS ice loss (Christianson et al., 2016; Jenkins et al., 2018). Figure 4 shows timeseries of zonal wind anomalies within the three magenta regions shown in Figure 2 during 1900-2020, the full time period of the paleoclimate reconstruction and ERA5 reanalysis. The regions extend zonally from 102-115 °W, and meridionally from 73-75 °S (shelf), 70.2-71.8 °S (shelf break), and 61-63 °S (deep

ocean). The statistics shown in each panel quantify the interannual correlation in zonal wind anomalies between the reconstruction and ERA5 during their period of overlap, showing that the reconstruction skill decreases towards the south (O'Connor et al., 2021). Consistent with the cyclonic trends in Figure 2a, over the 1920-2000 period the reconstruction features westerly trends in the deep ocean, no trend at the shelf break, and easterly trends over the shelf. However, there is substantial variability in all three regions on both interannual and interdecadal

timescales. Several large multi-annual anomalies are visible in the record, which overall appears to be dominated by interdecadal variability. Shelf break winds contain variability that reverses on a timescale of approximately 50 years (Figure 4b). We are not aware of any previous studies documenting this slow wind variability in this region, which may be crucial to the WAIS.

In principle, this wind variability arises through a combination of external forcing and internal climate variability. In common with the previous sections, the individual contributions can be separated by combining reconstructions and simulations. Figure 5 shows timeseries of the separate contributions to the zonal wind anomalies, for the three selected regions, over the historical and future time periods. The two columns both show the same annual timeseries in thin lines, but then emphasise either interdecadal evolution (left) or centennial trends (right) in thick

lines. The total externally-forced zonal wind anomalies for each location are the LENS ensemble mean. Historical GHG-forced and ozone-forced anomalies are the LENS ensemble mean minus the XGHG and XOZO ensemble means respectively. The internally-generated anomalies are the reconstructed anomalies minus the externally-forced anomalies (the LENS ensemble mean).

Significant externally-generated westerly wind changes occur over the deep ocean and shelf break during the historical period (Figures 5a-d, black lines). Over the deep ocean, GHG-forced changes are established from approximately 1960 onwards, while ozone-induced changes increase rapidly after approximately 1980 (Figure 5a, blue and green lines). This leads to significant historical GHG- and ozone-forced trends (Figure 5b). As noted above, the shelf break region sits in a transition zone where the forced responses are much weaker (Figure 5c).

While the overall externally-forced trend is significant, neither the GHG-forced or ozone-forced trends are significant individually (Figure 5d). There are no significant externally-forced changes over the shelf region (Figures 5e-f). All of these results are consistent with the spatial patterns in Figure 2, since the derivation of the trends and significance test are identical.

The internally-generated variability is substantial in all regions and shows many important features (Figure 5, red lines). Annual anomalies (thin red lines) are large in amplitude relative to the externally-forced changes. A particularly large westerly wind anomaly occurred over the shelf and shelf break in 1940 (Figures 5c and 5e), which is clearly part of an anticyclonic feature as it is accompanied by an easterly anomaly over the deep ocean



to the north (Figure 5a). The exceptional strength of this climatic anomaly is well known from ice cores (Schneider
and Steig, 2008; Steig et al., 2013), which are constraining the reconstruction here. Interdecadal anomalies (thick
red lines, left column) also show substantial variability. This variability broadly shows a 50-year reversal, with
opposing easterly and westerly anomalies between the deep ocean and shelf break. Internally-generated easterly
trends (right column) are also large, and play an important role in counteracting the externally-forced westerly
trends. At the shelf break the externally-forced trend is cancelled completely by the internally-generated trend,
while in the deep ocean approximately half of the externally-forced trend is cancelled. On the shelf, all variability
and trends are internally generated.

Given the known importance of Pacific variability to this region, Figure 6 examines the extent to which this
internal variability is related to the IPO tripole index. The IPO emerges only on long timescales (Newman et al.,
2016), so the tripole index is usually calculated from monthly SSTs, and then subjected to a 13-year filter to yield
the IPO. In Figure 6, we plot the standardised annual-mean and 13-year mean time series of both the IPO index
and the reconstructed internal variability in zonal winds. The statistics of the annual-mean and 13-year mean
correlations are shown in each panel. For the deep ocean region the correlations are negative, so the IPO index is
reversed in Figure 6a for illustrative purposes.


Annual anomalies in reconstructed internal variability are significantly correlated to the annual-mean values of
the IPO index in all regions (Figure 6). This is remarkable considering the independent origin of these datasets.
The IPO index is obtained from optimal interpolation of historical ocean temperature measurements (Rayner et
al., 2003), while the internal variability in winds is derived by combining climate model simulations and a
reconstruction that uses proxy records from ice cores, tree rings, and coral records. The most extreme annual
events in the record, around 1940 and 2000, seem very well explained by the tropical Pacific. The reversing sign
of the correlations between shelf and deep ocean illustrates the fact that a positive ENSO or IPO anomaly is
associated with anticyclonic wind anomalies associated with a local high pressure anomaly (Holland et al., 2019).

Unfortunately, there are insufficient data to assess the linkage on longer timescales. 13-year mean internally-
generated wind anomalies are not significantly correlated to the IPO (Figure 6), which is unsurprising given that
there are only ~70 years of data, and there appears to be a ~50-year reversal within the records. The change in
sign between shelf and deep ocean correlations appears to persist on 13-year timescales, suggesting the local
pressure anomaly pattern. On centennial timescales this reversal disappears, with all internally-generated trends
being easterly (Figures 2c and 5b, 5d, 5f). We know very little about the characteristics of internal variability on
centennial timescales. As noted above, the internally-generated centennial trends do not follow the IPO pattern
around the tropics (Figure 1), but that does not preclude the sub-centennial variability from being strongly
influenced by the IPO. Similar results are obtained with an alternative paleoclimate reconstruction (Q. Dalaiden,
personal communication, 2022).


Holland et al. (2019) found a significant historical westerly wind trend at the shelf break, but the same shelf-break
box is found to have no zonal wind trend in the present study (Figure 4b). These studies consider the same
externally-forced trends (from LENS), but differ in their consideration of internally-generated trends. Holland et



al. (2019) used the Pacific Pacemaker (PACE) ensemble mean to estimate the role of internal variability associated

with the tropical Pacific, and were unable to constrain variability from other sources. The present study uses the paleoclimate reconstruction to estimate the role of variability from all sources, Pacific and otherwise. In the PACE ensemble mean the Pacific variability enhances the externally-forced westerly trend (Holland et al., 2019), while in the reconstruction the internal variability cancels the externally-forced trend (Figure 5d). On further investigation, we found that the internally-generated trends in the reconstruction sit within the PACE ensemble

spread, but do not follow the PACE ensemble mean. We conclude that additional non-tropical-Pacific variability is captured in the reconstruction that modifies the mean tropical Pacific trend captured by PACE. This is supported by the fact that the reconstructed trends deviate significantly from a classical Pacific pattern (Figure 1). This is a further illustration of the value in using the paleoclimate reconstruction to estimate the real historical trajectory of internal variability. While the details of this study differ from Holland et al. (2019) within the shelf-break box, the

present results support the overall conclusion of that study that the region is impacted by westerly wind trends with a substantial anthropogenic component, modulated by internal variability on all timescales.

In summary, the paleoclimate reconstruction indicates that zonal winds are subject to strong variability on all timescales. Centennial trends feature a cyclonic pattern that switches from westerly over the deep ocean to easterly

on the shelf (Figure 4). The variability is strong on interannual timescales, and also expressed on interdecadal timescales, with a ~50 year reversal. Wind anomalies are driven by both external forcing and internal variability (Figure 5). The externally-forced part has distinct GHG and ozone signatures. The internally-generated component is partly related to the tropical Pacific, which induces variability on interannual and interdecadal timescales (Figure 6).

**3.2 Future changes**

We next consider the 'future' period for which consistent projections are available, 2001-2080. These analyses are necessarily different as we cannot use the paleoclimate reconstructions. We first consider externally-forced changes. Figure 7a shows the externally-forced wind trends following RCP8.5 (i.e. the LENS ensemble mean), and Figures 7b-d show the other scenarios. Historical and future responses are directly comparable in Figures 2

and 7 as the plotting is identical.

High (RCP8.5) and intermediate (RCP4.5) forcing scenarios (Figures 7a and 7b) feature strong externally-forced westerly wind trends over most of the Southern Ocean (Bracegirdle et al., 2020; Goyal et al., 2021) but much weaker trends over the Amundsen Sea shelf (Bracegirdle et al., 2014). Weaker forcing scenarios are cast in the

context of the Paris Agreement, which commits nations to "holding the increase in the global average temperature to well below 2°C above pre-industrial levels and pursuing efforts to limit the temperature increase to 1.5°C". The 2.0degC scenario substantially reduces future wind trends in this region (Figure 7c), while 1.5degC eliminates the westerly trends entirely (Figure 7d). Stabilising global warming removes the equator-to-pole gradient in temperature trends that drives the westerly wind trends. We speculate that the small remaining wind trends are

likely to be very sensitive to the forcing pathway by which temperatures are stabilised. The 1.5degC scenario involves aggressive emissions mitigation and a rapid transition to negative carbon emissions (Sanderson et al., 2017). Historical GHG emissions commit Earth's climate to warming and wind changes for most of the 'future'



period studied, so extreme emissions changes are required to reverse these by 2080. Even then, the 1.5degC scenario merely prevents any future trends, and does not reverse the historical changes.


As with the historical changes, we may combine ensembles to derive the role of individual external forcings. Figure 7e shows the GHG-forced changes in the RCP8.5 scenario (i.e. LENS minus XGHG). This future GHG response is approximately twice as strong as the historical GHG response (Figure 2d), with a similar pattern. We cannot directly determine future ozone-forced changes because the fixed-ozone ensemble XOZO terminates in

2005 (Table 1). However, Figure 7f shows the combined response to all non-GHG external forcings (XGHG minus PIctrl), which we are confident is dominated by the ozone response. For the historical period the non-GHG response closely resembles the ozone response, and in future scenarios the influence of external forcings other than GHGs and ozone (e.g. aerosols) are even weaker (van Vuuren et al., 2011). The future non-GHG response (Figure 7f) shows a weakening of the westerlies, as expected from the prescribed recovery of stratospheric ozone

(Son et al., 2010; Sigmond et al., 2011; Barnes et al., 2014), and in almost exact opposition to the historical ozone-induced changes (Figure 2f). Thus the models yield the expected result that the future westerly wind trends are driven by GHGs, but partially compensated by ozone recovery (Thompson et al., 2011). The influence of ozone recovery is expected to be very similar between forcing scenarios (Keeble et al., 2021), so the difference between scenarios in Figures 7a-7d is determined by the extent to which increasing GHGs outweigh this recovery.


The future evolution of the different components of the externally-forced wind changes is shown in Figure 5, which again shows results for RCP8.5. In this strong forcing scenario, future externally-forced trends are of similar magnitude to historical externally-forced trends in all regions. Over the deep ocean (Figure 5b), an increase in the GHG-forced westerly wind trend is compensated by a negative westerly trend from stratospheric ozone recovery

(represented by the non-GHG response). Over the shelf break (Figure 5d), the ozone-forced trend remains insignificant, but the GHG-forced trend becomes significant in the future period. The recovery of ozone-forced wind trends is focussed earlier in the 'future' period (Keeble et al., 2021), so the overall externally-forced trends are weaker during this period and stronger after (Figure 5a).

Internal variability was an important contributor to wind trends over the Amundsen Sea in the past (Figures 2 and 5), and could also be important in future. The evidence suggests that tropical Pacific variability is primarily natural in origin (Stevenson et al., 2012; Cai et al., 2015; Yeh et al., 2018), and cannot be predicted on timescales longer than a few years (Lou et al., 2019). Furthermore, centennial variability in this region does not even appear to follow the recognised modes of tropical Pacific variability (Figure 1). Therefore, the role of internal variability

places a substantial and potentially irreducible uncertainty on any projection of future wind changes over the Amundsen Sea, particularly in the shelf region.

One way to quantify the influence of internal variability on future wind trends is by considering the ensemble spread in the projections. Figure 8a shows the LENS ensemble-mean trends (i.e. the RCP8.5 externally-forced

trends in Figure 7a). Figure 8b quantifies the associated ensemble spread, plotted as the wind trends associated with +1 standard deviation in both zonal and meridional wind trends. This figure shows that there is substantial intra-ensemble variance in the trends, which is evenly spatially distributed and has no obvious preferential wind



direction. Figures 8c and 8e show the future wind trends in two LENS members, chosen manually to illustrate the range of intra-ensemble variance. Both members have strong westerly wind trends overall, but differ in their trends over the Amundsen Sea shelf. Figures 8d and 8f isolate the differing contribution of internal variability in these two members, by subtracting the LENS ensemble mean (the externally-forced trends) from both. The internally-generated trends have a centre of action north of the Amundsen Sea shelf in both cases, but the trends have either a cyclonic or anticyclonic orientation depending upon the particular trajectory of internal variability manifested in each simulation.

Another way to illustrate the potential role of future internal variability is to simply imagine that the historical variability could be repeated, or exactly reversed. The advantage of this approach is that our knowledge of the historical variability is constrained by observations. Comparing the historical internally-generated trends (Figure 2c) to the LENS ensemble spread of future internally-generated trends (Figures 8b, 8d and 8f) shows that this is a representative illustration. Following this rationale, Figure 9 illustrates possible future wind trends that combine forcing scenario uncertainty with the historically-based estimate of future internal variability. This is illustrated simply by adding or subtracting the historical internally-generated trends (Figure 2c) to the externally-forced trends in the 1.5degC and RCP8.5 forcing scenarios (Figures 7d and 7a). It is clear that external forcing and its scenario uncertainty controls westerly trends over the deep ocean, while internal variability and its irreducible uncertainty controls wind trends over the Amundsen Sea shelf and shelf break.

## 4 Discussion

Recent wind anomalies are known to have driven variability in Amundsen Sea ocean conditions and ice-shelf melting that have influenced ice loss from the WAIS (Thoma et al., 2008; Dutrieux et al., 2014; Christianson et al., 2016; Jenkins et al., 2018). Reconstructed winds show that even larger anomalies occurred in the past (Figure 4), so it is reasonable to assume that these anomalies also induced an ice-sheet response. The reconstructions also reveal interdecadal (~50-year) variability with an amplitude approximately half that of the annual anomalies. Since ice sheets respond more fully to slower variability (Williams et al., 2012; Snow et al., 2017; Hoffman et al., 2019), it is very likely that these interdecadal anomalies have influenced the WAIS.

On centennial time scales, the reconstruction shows that strong westerly wind trends were prevalent over most of the Southern Ocean (Figure 2). These overall trends were driven by external forcing but partly compensated by internally-generated trends. Over the Amundsen Sea shelf break these contributions cancelled, leaving no significant overall trend, in contrast to earlier results based on climate models alone (Holland et al., 2019). Over the Amundsen Sea shelf, externally-forced trends were absent entirely and an internally-generated easterly trend occurred. Where present, the compensating externally-forced and internally-generated trends are both of similar magnitude to the recent wind variability that we know to be influential over WAIS ice loss. This implies that both the trends, and their compensation, have been important to the historical evolution of the ice loss.

This evidence suggests a possible narrative for the historical changes in the WAIS. Sediment records show that the ungrounding of Pine Island Glacier commenced in the 1940s (Smith et al., 2017), when the reconstruction shows extreme annual wind events within a multi-decadal period of anomalously anticyclonic winds (Figure 4).



These anomalies were all internally generated (Figure 5) and partly associated with tropical Pacific variability (Figure 6). By the mid-1970s the ice shelf was fully ungrounded (Jenkins et al., 2010; Smith et al., 2017), following a period of retreat that was probably enhanced by ice and ocean feedbacks (De Rydt et al., 2014; De

Rydt and Gudmundsson, 2016). The triggering internal variability had reversed by the 1960s, but by then the GHG-forced wind anomalies had appeared (Figure 5). Perhaps, without external forcing, the reversed internal variability might have allowed the ice to re-advance and fully re-ground on the ridge. In the 1980s the external forcing increased further with ever-increasing GHGs and the onset of ozone depletion, and the internally-generated anomalies became anticyclonic again. The wind changes resulting from these anthropogenic and natural

drivers were of approximately equal magnitude, suggesting that a combination of these factors drove the current period of rapid, accelerating ice loss (Mouginot et al., 2014; Konrad et al., 2017; Shepherd et al., 2019).

While much remains uncertain, this narrative can simultaneously satisfy lines of evidence from paleoclimate proxies, Amundsen Sea sediment records, ocean observations, climate model simulations, and the satellite record

of ice loss. Importantly, it also resolves an apparent paradox. The WAIS is thought to have been broadly stable for ~10,000 years in this region (Larter et al., 2014) before the current retreat was initiated in the 1940s (Smith et al., 2017) by internal climate variability (Figures 4 and 5). However, by definition it is very unlikely that this natural variability was unprecedented in the previous millennia, so why did it trigger an exceptional ice retreat? One way to resolve this apparent paradox is if the initial retreat was natural, but the subsequent failure of the ice

sheet to re-advance was influenced by anthropogenic forcing. Perhaps the ice sheet experienced several short-lived retreats during the last few millennia, from which it recovered, but it failed to recover from the 1940s retreat because external forcing over-rode the subsequent reversal of the natural climatic variability (Figures 4 and 5). We emphasise that this is not the only possible resolution of this 'paradox', however. For example, if some slow background change had occurred over the millennia, such as a reduction in surface accumulation, this could have

gradually reduced the stability of the glaciers in the region. The 1940s variability may then have been able to destabilise the glaciers simply because they were more vulnerable.

Turning to the future changes, 2000-2080, we again find that both external forcing and internal variability are of comparable importance. Remarkably, the choice of emissions scenario is important to wind trends everywhere

apart from the Amundsen Sea shelf. On the shelf, future wind trends are dictated solely by internal variability, while over the deep ocean the external forcing plays a larger role. Under an aggressive emissions mitigation scenario that limits global warming to 1.5°C, there are no externally-forced zonal wind trends during the future period. Under high (RCP8.5) or even intermediate (RCP4.5) emissions scenarios, future externally-forced zonal wind trends approximately equal the historical externally-forced trends. Thus, future forcing will range between

maintaining the existing externally-forced changes (1.5degC scenario) and approximately doubling them (RCP8.5). To the extent that ice loss from the WAIS is driven by externally-forced wind trends, that part of the ice loss cannot be reversed before 2080 and will probably increase.

Internally-generated trends provided an important compensation of the externally-forced trends during the

historical period, and could be similarly influential in future. If externally-forced trends were eliminated in future, internally-generated trends could dictate the future trajectory of the WAIS (Figure 9). On the other hand, if higher



emissions scenarios are followed, internal variability may do little more than influence the timing and rate of the ongoing ice loss. Minimising future wind-driven WAIS ice loss will require strong emissions mitigation, but the uncontrollable future influence of internal climate variability could be equally important.


There are, of course, many caveats to these findings. This study considers a single paleoclimate reconstruction and climate model, and is therefore subject to the structural uncertainty inherent in these sources (see section 2). The study also only considers wind forcing of changes in the ocean, and thermodynamic forcings may also be important. Ice sheet dynamics and ice/ocean feedbacks certainly modulate the ice response to any climatic forcing.

We urgently require further information about the oceanographic implications of the wind changes found in this study. However, we emphasise that all of the wind changes discussed are comparable in magnitude to the recent variability, which we know to have been influential over WAIS ice loss.

## 5 Conclusions

Over recent decades, wind-driven variability in the Amundsen Sea has regulated ocean melting of the WAIS. This
study combines paleoclimate reconstructions and climate model simulations to understand wind changes over the Amundsen Sea during the 20th and 21st centuries.

By combining these two sources of information, we are able to separate natural internally-generated wind changes from anthropogenic externally-forced changes. Both are important. Firstly, internal variability on interannual and
interdecadal time scales has a comparable magnitude to centennial trends. Secondly, the centennial trends themselves are generated by comparable contributions from external forcing and internal variability. To the extent that wind-driven changes control WAIS ice loss, both external forcing and internal variability are important contributors.

Historical wind trends (1920-2000) have two components: acceleration of the westerlies over the deep ocean, forced by GHGs and ozone depletion, and an internally-generated cyclonic trend pattern centred over the Amundsen Sea. Over most of the region, internally-generated easterly wind trends compensate externally-forced westerly wind trends. Historical winds also exhibit strong variability, linked to the tropical Pacific, including both strong interannual anomalies and also interdecadal variability that reverses on a timescale of approximately 50
years.

This evidence suggests a possible narrative for historical ice loss from the WAIS. Ice retreat was triggered in the 1940s by internal variability. This variability had reversed by the 1960s, but by then GHG-forced wind changes had started to increase. Perhaps without external forcing, the reversed internal variability may have allowed the
ice sheet to re-advance. In the 1980s the external forcing increased further with the onset of ozone depletion, and the internal variability changed sign again. These changes drove the current period of rapid, accelerating ice loss. There remain many uncertainties with this narrative, but it resolves an apparent 'paradox' whereby the present ice loss appears to have been triggered naturally, but involves a retreat unprecedented in millennia. Perhaps short-lived retreats have occurred several times in the past, but the ice always re-advanced in the absence of
anthropogenic forcing.



During the future period (2000-2080), westerly wind trends driven by GHGs are partially offset by stratospheric ozone recovery. Wind trends are responsive to forcing scenario, but only extreme emissions mitigation consistent with a Paris Agreement target of 1.5°C warming above pre-industrial is able to prevent further wind trends during

this period (let alone offset historical trends). The choice of anthropogenic forcing scenario is most influential on westerly trends over the deep ocean, while the irreducible uncertainty associated with internally-generated variability is strongest on the Amundsen Sea shelf. The minimal future wind-driven WAIS ice loss will require strong emissions mitigation and a favourable evolution of natural climate variability.

**Author Contributions**

PH and GKO conceived the study and led the data processing. All authors discussed the results and implications and collaborated on writing the manuscript at all stages.

**Acknowledgements**

We are grateful to the originators of the many open-access datasets synthesised in this study, particularly the PAGES2K paleoclimate proxy database and the CESM1 climate model simulations. We thank all the scientists,

software engineers, and administrators who contributed to the development of CESM1, which is primarily supported by the National Science Foundation (NSF). GKO was supported by the NSF Graduate Research Fellowship Program. KN was supported by award NE/S011994/1. DPS was partially supported by NSF grant 1952199, and partially supported by the National Center for Atmospheric Research (NCAR), which is a major facility sponsored by the NSF under Cooperative Agreement no. 1852977.

**Data Availability**

The CESM1 simulations are available at the NCAR Climate Data Gateway, as detailed in the references in Table 1. The paleoclimate reconstruction are available through O'Connor et al. (2021).

**Competing Interests**

The authors declare no competing interests.

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



| Name | Description | Years | n | Reference |
|---|---|---|---|---|
| LENS | Large Ensemble. Initial-condition ensemble of historical and projection simulations under all external forcings, following RCP8.5 scenario for projections. | 1920-2100 | 40 | Kay et al. (2015) |
| PIctrl | Pre-industrial control ensemble. 1,760 years of control simulation treated as ensemble of 22 non-overlapping 80-year simulations, used for both historical and projection periods. | 80 (see left) | 22 | Kay et al. (2015) |
| PACE | Pacific Pacemaker ensemble. Same as LENS historical, but with tropical Pacific SST anomalies constrained to follow observations. | 1920-2013 | 20 | Schneider and Deser (2018) |
| MENS | Medium Ensemble. Same as LENS projections, but following RCP4.5 forcing scenario. | 2006-2080 | 15 | Sanderson et al. (2018) |
| 2.0degC | Same as LENS projections, but following a forcing scenario designed to keep global-mean temperatures below 2°C above pre-industrial. | 2006-2100 | 10 | Sanderson et al. (2017) |
| 1.5degC | Same as LENS projections, but following a forcing scenario designed to keep global-mean temperatures below 1.5°C above pre-industrial. | 2006-2100 | 10 | Sanderson et al. (2017) |
| XOZO | No-Ozone-Depletion ensemble. As LENS, but with stratospheric ozone fixed after 1955. | 1920-2005 | 8 | England et al. (2016) |
| XGHG | No-GHG ensemble. As LENS, but with atmospheric GHGs fixed at 1920. | 1920-2080 | 20 | Deser et al. (2020) |
| XIND | No-Industrial-Aerosol ensemble. As LENS, but with industrial aerosols fixed at 1920. | 1920-2080 | 20 | Deser et al. (2020) |
| XBMB | No-Biomass-Burning ensemble. As LENS, but with biomass-burning aerosols fixed at 1920. | 1920-2029 | 15 | Deser et al. (2020) |

**Table 1: Description of the CESM1 model ensembles considered in this study.**

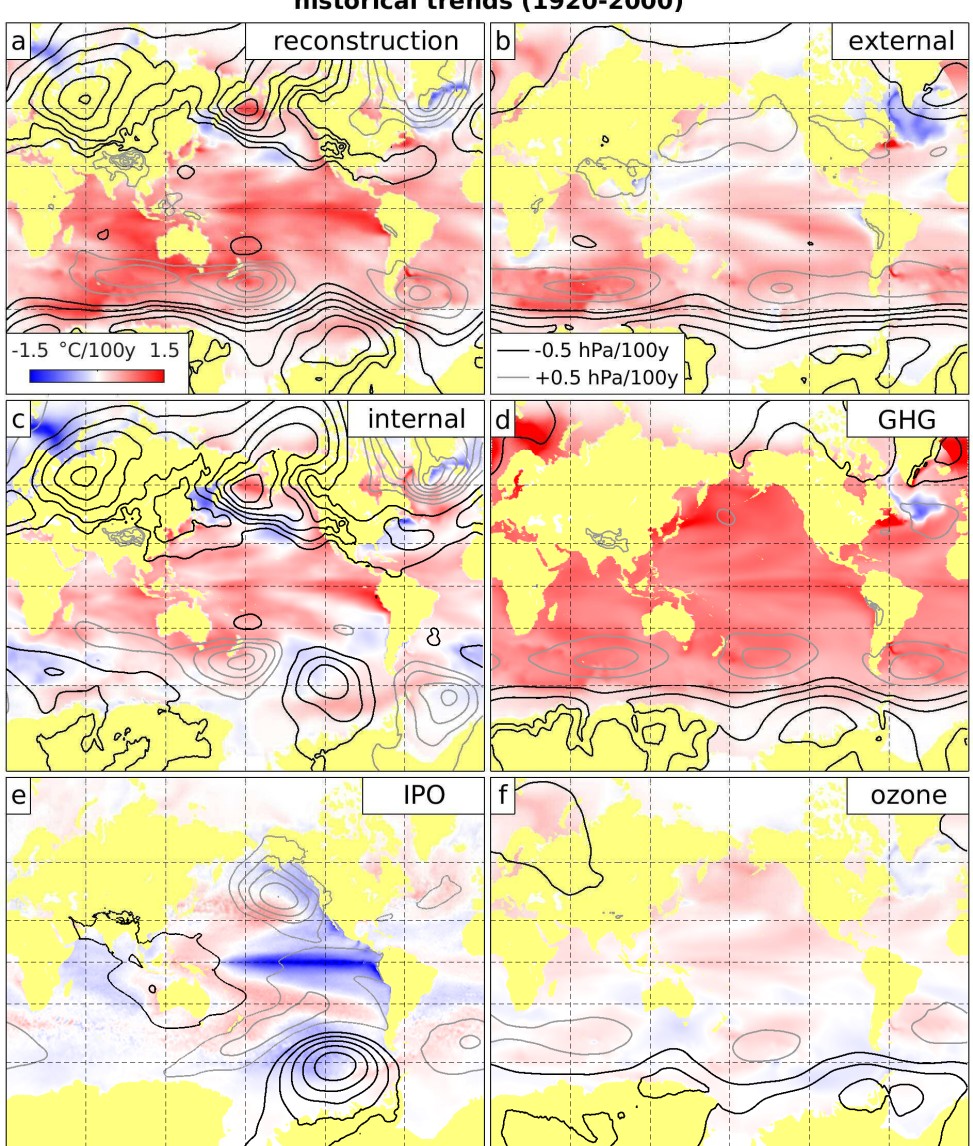

**Figure 1: Global trends in sea-surface temperature (colour) and sea-level pressure (contours) over the historical period.**
**For SLP, positive and negative contours are plotted in grey and black respectively, and the zero contour is omitted. a)**
**Trends in the paleoclimate reconstruction. b) Trends caused by external forcing. c) Trends associated with internal**
**variability (panel a minus panel b). d) and f) Trends caused by greenhouse-gas forcing and ozone depletion,**
**respectively. e) Trends associated with a hypothetical trend of -1°C/century in the Interdecadal Pacific Oscillation**
**index.**
### historical wind trends (1920-2000)

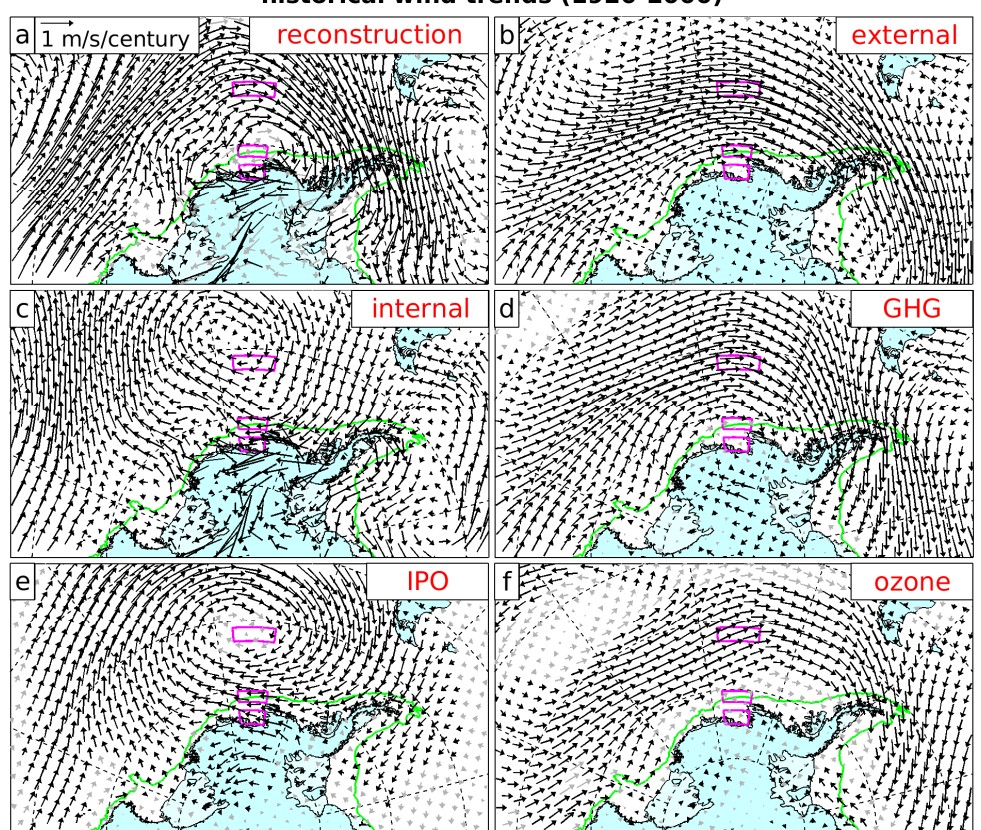

**Figure 2: Historical trends in near-surface winds over West Antarctica and the South Pacific.** The six panels are analogous to Figure 1, but with the analysis repeated for near-surface wind trends rather than SLP and SST, and focussing on the Pacific sector of the Southern Ocean. Black vectors have a zonal or meridional wind trend significant at the 95% confidence level. The significance test is different for the different panels (see main text). Green contour is the 1000-m isobath at the continental shelf break. Magenta boxes show three selected regions of interest, including (south to north) the Amundsen Sea shelf, shelf break, and deep ocean. These are the locations of timeseries in Figures 4-6.

**historical wind trends (1920-2000)**

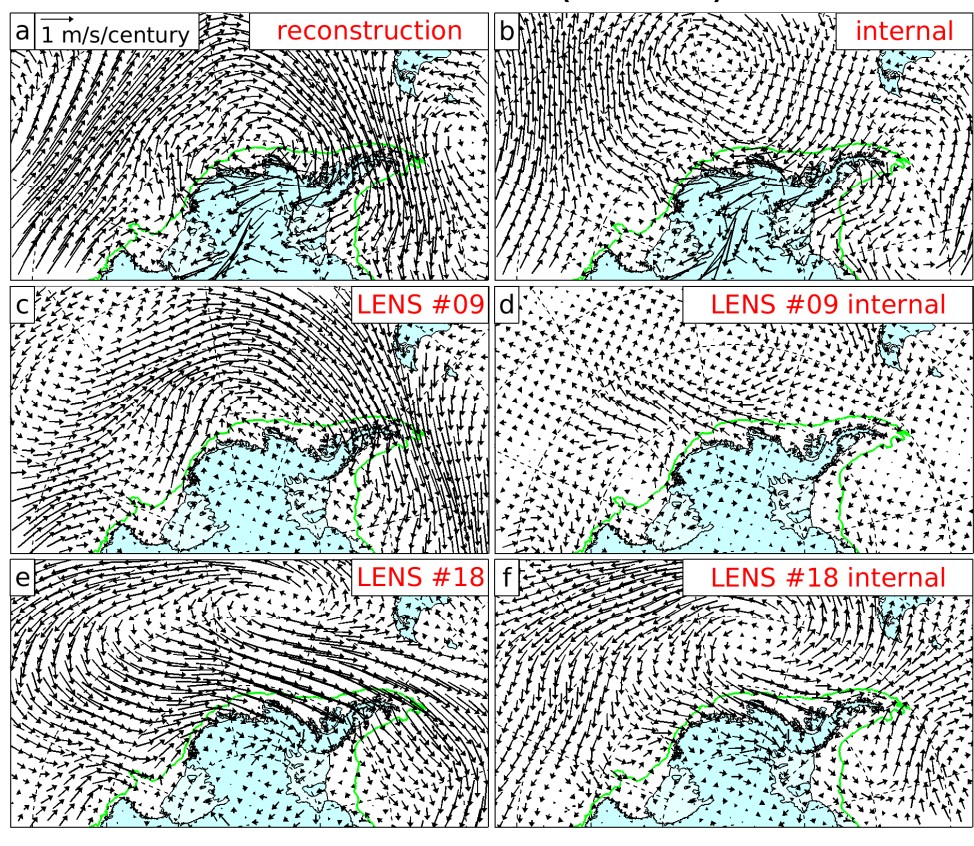


**Figure 3: Comparison between paleoclimate reconstruction and selected LENS ensemble members. a) Reconstructed historical wind trends. c) and e) Wind trends in selected illustrative ensemble members. b), d) and f) Deviation of each of these fields from the externally-forced trends in the LENS ensemble mean, illustrating the role of internal climate variability in each field.**

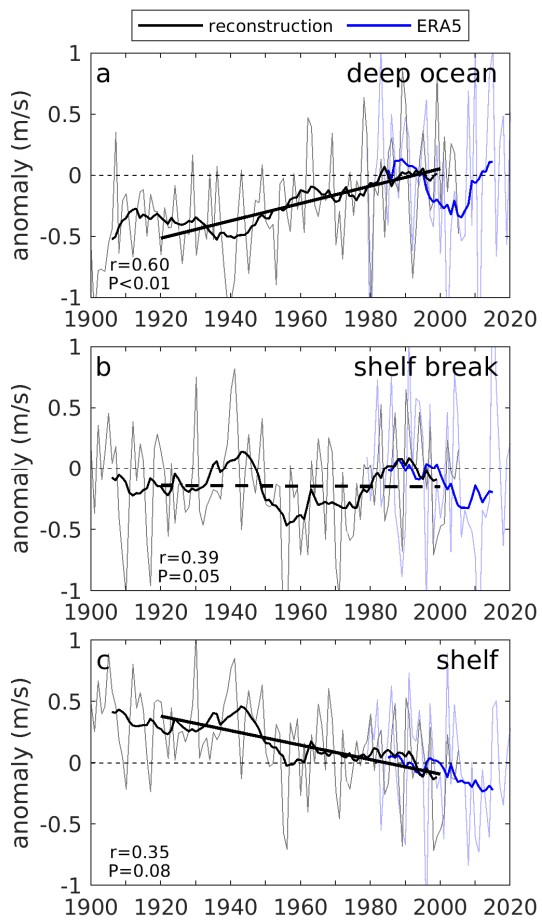

**Figure 4: Historical evolution of zonal wind anomalies during 1900-2020. Time series are calculated in the three regions highlighted in Figure 1 from the paleoclimate reconstruction and ERA5 reanalysis. Thin lines show annual anomalies, while thick lines show the 13-year running mean, selected because it is characteristic of the Interdecadal Pacific Oscillation. 1920-2000 trend lines are shown solid if the trend is significant at the 95% confidence level and dashed otherwise. Anomalies are plotted relative to mean values over 1979-2005, the period of overlap between the reconstruction and ERA5. Statistics are given for the correlation between datasets during this overlap.**



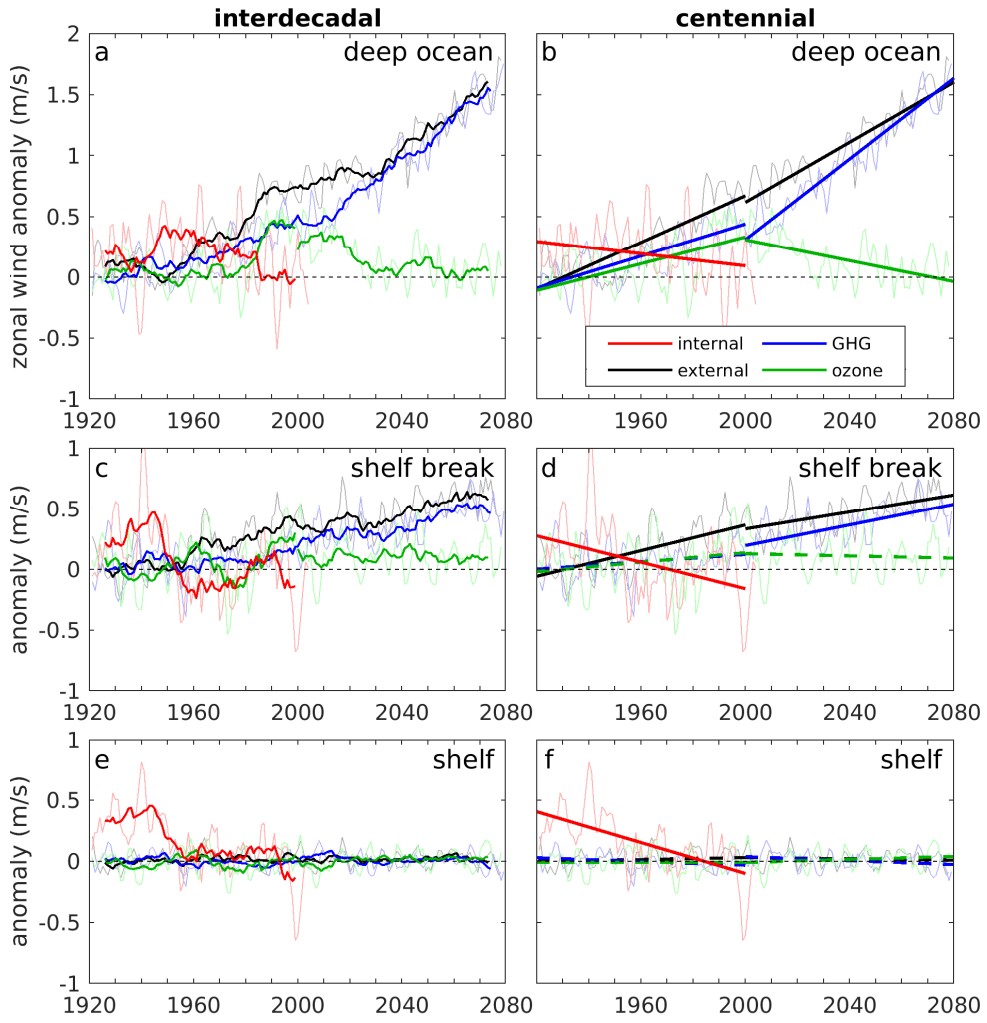

**Figure 5: Historical and future influences on zonal wind anomalies.** Panels show the internally-generated, total externally-forced, and GHG- and ozone-forced contributions to wind changes in the three regions shown in Figure 2. Future changes are shown under the strong forcing RCP8.5 scenario. All panels show annual-mean timeseries (thin lines). The left column also shows the same timeseries after a 13-year running mean (thick lines) to highlight the interdecadal evolution of the responses. The right column also shows historical and future trends, plotted solid if the trend is significant at the 95% confidence level (see text) and dashed otherwise. Ozone is plotted as the ozone response before 2000 and the non-GHG response afterwards. Internally-generated anomalies are plotted relative to 1979-2005, as in Figure 4.

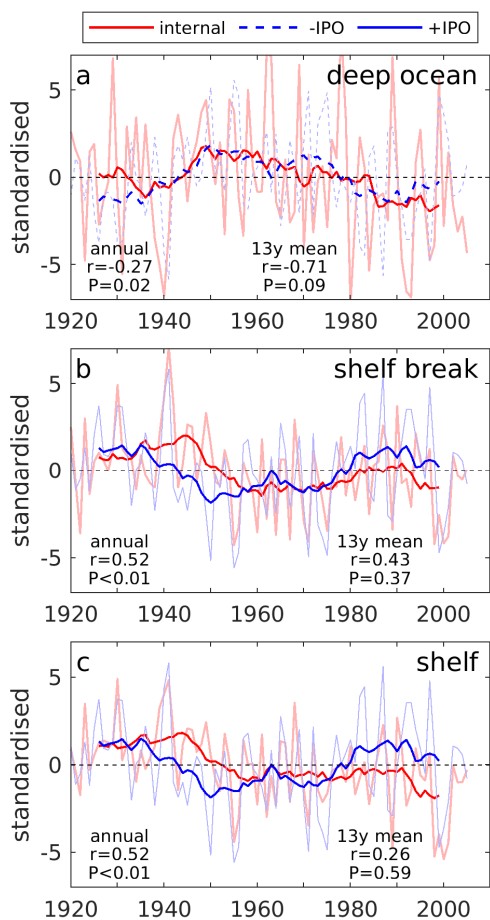

**Figure 6: Evolution of internally-generated zonal wind anomalies and their relationship to the Interdecadal Pacific Oscillation. Panels show the internally-generated anomalies and IPO index in the three regions shown in Figure 2. Thin lines show annual anomalies, while thick lines show a 13-year running mean. All time series are standardised by the 13-year-running-mean data. The IPO index is inverted in panel a for illustrative purposes because its correlation to wind anomalies is negative in the deep ocean region. Statistics show the correlation between the two time series on annual and 13-year timescales.**

**future wind trends (2000-2080)**

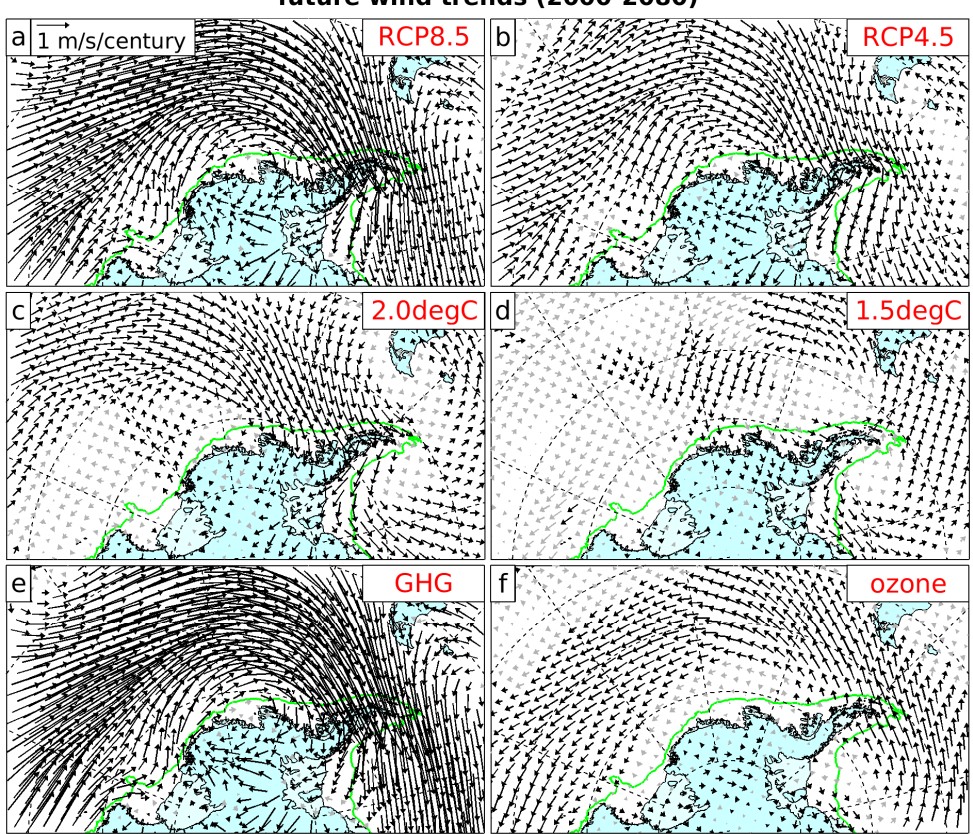

**Figure 7: Projected wind trends over West Antarctica and their attribution. Panels a-d show externally-forced wind trends in four different forcing scenarios, progressing from the strongest anthropogenic forcing (RCP8.5) to an aggressive emissions mitigation scenario (1.5degC). Panels e and f show the individual contributions of greenhouse gases and ozone depletion in the RCP8.5 scenario.**
## future wind trends (2000-2080)

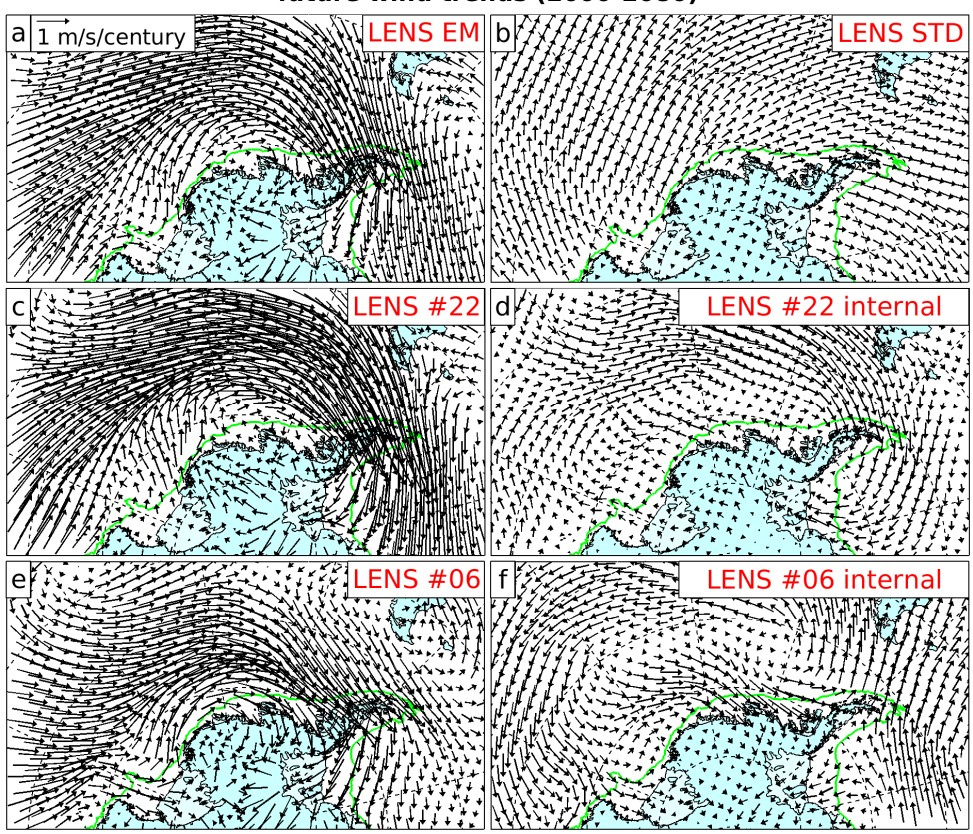

**Figure 8: Influence of internal variability on projected wind trends over West Antarctica. a) LENS ensemble-mean wind trends (the RCP8.5 externally-forced trend in Figure 7a). b) LENS intra-ensemble standard deviation in wind trends, plotted as vectors illustrating one positive standard deviation in both zonal and meridional trends. c) and e) Wind trends in selected illustrative LENS members. d) and f) Deviation of the trends in these selected members from the ensemble-mean trend, illustrating the role of internal climate variability.**

**future wind trends (2000-2080)**

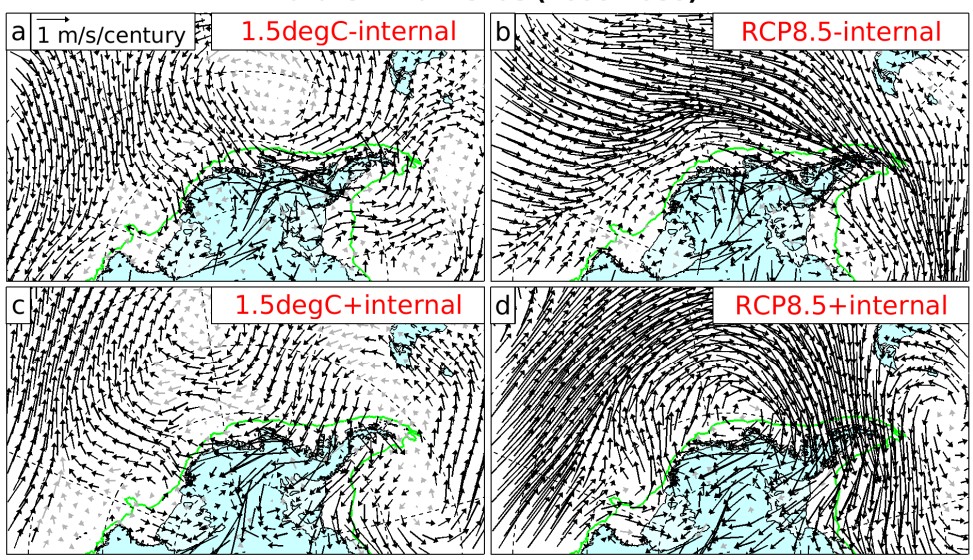

**Figure 9: Illustrative wind trend projections under different combinations of external forcing scenario and a single realisation of internal variability. The columns denote wind trends projected under weak (left) and strong (right) external forcing, while the rows denote the influence of either subtracting (top) or adding (bottom) the wind trends associated with historical internal variability, which is used for illustrative purposes. The projected externally-forced wind trends are shown in Figures 7a and 7d, while the historical internally-generated wind trends are shown in Figure 915   2c.**