# Peer review of "Anthropogenic and internal drivers of wind changes over the Amundsen Sea, West Antarctica, during the 20th and 21st centuries"

_The Cryosphere, 2022_

## Author Comment (AC1)

Overview – this study investigates historical (1920-2000) and future (2000-2080) changes in near surface pressure and winds over the South Pacific using a combination of a spatially complete paleoclimate reconstruction and a large suite of climate model simulations.  It provides convincing evidence of the relative roles of internal variability and forced variability (from greenhouse gases and ozone depletion).  Importantly, the paper also provides a much-needed possible narrative for the roles of natural variability followed by external forcing to understand historical variations in ice loss from the WAIS since 1940s.

The paper is very well written, easy to follow, and the scientific analysis in my perspective is sound. The team is to be commended on this excellent study, which is well conceived and an important scientific advancement. The authors note many caveats to the present study, which I also appreciate.

My main concern is that the paper is primarily based on gridded model or paleoclimate-based reconstruction data, and does not incorporate the wealth of observations other than ERA5 after 1979. It would be helpful to see the agreement between the reconstruction at least and pressure observations (available across all SH midlatitudes since 1920, and Orcadas since 1903) to see at least the agreement in South America / South Atlantic.  For more complete investigation on the agreement (and for some aspects of the deepening of the ASL), comparisons could be made with Antarctic data after 1957 when trends seem quite large (making note to include the critical point of Byrd station in West Antarctica as another potential estimate of observed change near the Amundsen Sea apart from measurements along the Antarctic Peninsula).  I do feel the point observations comparisons with the reconstructions would help to understand changes in observations apart from paleoclimate data and climate model data, and would round out the paper well (1 more figure), and provide further validation for the reconstruction that is not provided in the preceeding O'Connor et al. paper.

O'Connor et al. (2021) validate their reconstruction against other paleoclimate reconstructions of the 20th century, modern reanalyses datasets (since 1979), and longer-term reanalyses (since 1900). The longer-term reanalyses are constrained primarily by the station data mentioned above. O'Connor et al. (2021) also show that their reconstructed SAM index compares favourably to that of Marshall (2003), which is based solely on station pressure observations from Antarctica and the sub-Antarctic regions since 1958.  Thus the reconstruction has been validated using the earlier observations suggested by the reviewer.  In any case, we see no immediate reason why the reconstruction's skill in fitting modern reanalysis data would not apply to earlier periods, since the proxy data underlying the reconstruction are uniformly available throughout the 20th century.

We agree that directly comparing the O'Connor et al. (2021) reconstruction to each of the station pressure records around Antarctica would add detailed insight into the quality of the reconstruction at those locations. However, the applicability of such a comparison to the present study is not clear because there are no long-term station data near the Amundsen Sea. It is not clear how far any direct misfit to remote stations such as Orcadas or South America should limit our confidence in reconstructed Amundsen Sea winds. (In the previous validations mentioned above, this issue is avoided by using the reanalysis model or SAM index to link such stations to the Amundsen Sea.) As a result we feel that a systematic comparison at station locations is certainly useful in general, but beyond the scope of the present paper.

To illustrate the issue of applicability of station data, we compared the reconstructed geostrophic winds used in the paper to geostrophic winds calculated from the surface pressure dataset of Fogt et

al. (2019), which is constructed by interpolating Antarctic station pressure observations. Figure R1 shows an extended version of Figure 4 from the paper, to which the derived Fogt et al. (2019) winds have been added. The Fogt-derived winds are positively correlated to ERA5 over the deep ocean, but do not provide a useful constraint over the shelf break or shelf. (We speculate that negative correlations over the shelf reflect the Amundsen Sea Low, whose pattern is not captured by the spatial kriging in the Fogt dataset.) Over the deep ocean since 1957, the O'Connor and Fogt reconstructed winds are correlated, which is a very encouraging result considering the independent origin of these datasets. The level of correlation is similar to the fit between the O'Connor and Marshall SAM indices during that period (O'Connor et al., 2021). Prior to 1957 the fit between O'Conor and Fogt timeseries is much worse, which is unsurprising because the few direct station observations available during that period are extremely remote from the Amundsen Sea.

It is clear from the comments of both reviewers that the validation of the reconstructed winds is an important concern and needs to be better explained in the paper. We propose to add a new paragraph of text to section 2.1 detailing the various validation tests carried out by O'Connor et al. (2021) and

[Figure]

Figure R1: Copy of Figure 4 from the paper, but also containing zonal wind anomalies calculated from the Fogt et al. (2019) spatial interpolation of station pressure data.

presenting our rationale as to why these analyses support the use of the reconstruction in our study. We also propose to add some new text to section 3.1.3 detailing the statistics of our comparison of the reconstruction to the Fogt et al. (2019) dataset for winds over the Amundsen Sea. This will respond to the reviewer's comment by adding further text to the paper detailing the following points: i) Station data have previously been used to constrain the reconstruction through their influence on longer reanalyses and the Marshall SAM index; ii) Direct comparison of the reconstruction to station records would be a valuable step but will not necessarily constrain winds over the Amundsen Sea; iii) A new comparison to Fogt-reconstructed winds provides further station-based validation of the reconstructed winds over the deep ocean since 1958.

Minor comments:

Abstract – would be ideal to clarify the ice loss was not in reference to sea ice, but the grounded ice sheet

We will revise the text where needed.

L255-260, Fig. 1f – I also suspect the response to ozone is weaker as it is seasonally varying (strongest in DJF at the surface), so the annual mean reduces this signal.

We will add this point.

Wondering what role incorrect sea ice trends in the model may play in both the historical and future simulations? The climate model tends to overestimate observed sea ice trends compared to observations. Importantly, the sea ice trends have been most pronounced in the Pacific sector in observations, which is the area of study, so it is possible that there could be some impact of this on the pressure and wind trends in the region from the model, especially in the model ensemble means. Can the authors comment on this potential error, where appropriate, in the paper?

Since the advent of continuous satellite observations in 1979, sea-ice trends have been focussed on the Pacific sector of the Southern Ocean as a result of internal variability. The trends are driven by a negative trend in the IPO since the 1980s (Meehl et al., 2016; Purich et al., 2016). Holland et al (2019) and Schneider and Deser (2018) show that that the CESM1 is able to accurately represent this pattern of internally-generated variability over this part of the South Pacific. So there is no additional cause for concern in this region.

Overall, however, Schneider and Deser (2018) show that historical CESM1 simulations do feature an unrealistic circum-Antarctic trend of sea-ice loss since 1979. This ice loss is associated with excessive ocean surface warming, suggesting that the model does not subduct heat efficiently into the Southern Ocean interior. Such ocean model biases are the reason we focus on winds in this study as a proxy for ocean history in the Amundsen Sea, rather than considering the ocean model results directly.

These sea ice and SST trend biases do not seem to heavily influence model winds. Since 1979, the CESM1 accurately represents trends in the Amundsen Sea Low (England et al., 2016), wind trends over the Amundsen Sea (Holland et al., 2019), and the pattern of pressure trends over the South Pacific (Schneider and Deser, 2018). Thus, we are not unduly concerned about the sea ice and SST biases.

It remains possible that the model has an excessive wind response to external forcing over the longer time period since 1920. This cannot be validated directly against observations, since the real wind history combines both externally-forced and internally-generated changes. We can only note that the CESM1 externally-forced wind trends are representative of the wider CMIP5 ensemble in this region

(Holland et al 2019), and the ensemble of CESM1 historical trends comfortably includes the reconstructed historical trends (figure 3).

Model biases are discussed from line 166 onwards, including mention of previous work on the model's ability to represent sea-ice trends during recent decades (Schneider and Deser, 2018), so we will expand that discussion to include the relevant points: i) Overall, the CESM1 does contain biased sea ice trends in recent decades; ii) There is no additional cause for concern in the region of interest to this study; iii) The sea ice biases do not appear to be associated with wind biases; iv) The model is not an outlier in this regard.

L595-603 – really appreciate mentioning the caveats to the study. I think it is also important to mention that the study masks seasonal variability, limited by the paleoclimate reconstruction, that is important for tropical teleconnections (i.e., internal variability) and the role of ozone forcing.

We will add this point.

**References**

Fogt, R. L., Schneider, D. P., Goergens, C. A., Jones, J. M., Clark, L. N., and Garberoglio, M. J.: Seasonal Antarctic pressure variability during the twentieth century from spatially complete reconstructions and CAM5 simulations, Climate Dynamics, 53, 1435–1452, 10.1007/s00382-019-04674-8, 2019.

Marshall, G. J.: Trends in the Southern Annular Mode from observations and reanalyses, Journal of Climate, 16, 4134-4143, 2003.

Meehl, G. A., Arblaster, J. M., Bitz, C. M., Chung, C. T. Y., and Teng, H. Y.: Antarctic sea-ice expansion between 2000 and 2014 driven by tropical Pacific decadal climate variability, Nature Geoscience, 9, 590-+, 10.1038/Ngeo2751, 2016.

O'Connor, G. K., Steig, E. J., and Hakim, G. J.: Strengthening Southern Hemisphere westerlies and Amundsen Sea Low deepening over the 20th century revealed by proxy-data assimilation, Geophysical Research Letters, 48, e2021GL095999, 10.1029/2021GL095999, 2021.

Purich, A., England, M. H., Cai, W., Chikamoto, Y., Timmermann, A., Fyfe, J. C., Frankcombe, L., Meehl, G. A., and Arblaster, J. M.: Tropical Pacific SST Drivers of Recent Antarctic Sea Ice Trends, Journal of Climate, 29, 8931-8948, 10.1175/Jcli-D-16-0440.1, 2016.

---

## Author Comment (AC2)

**Quentin Dalaiden**

The study led by Paul Holland is very interesting and presents a heavy load of work on historical and future changes in surface winds in the Amundsen Sea area. The authors provide a clear quantification of the internal and external contributions to wind changes in this region where very large ice losses have been observed over the past decades. They also provide a narrative on the ice loss from the WAIS starting in the 1940s. This study is thus highly suitable for publication in The Cryosphere. I really enjoyed reading the paper, which I find very well written. Here are some comments:

Lines 54-55: Could you give more detail on the ice-ocean feedbacks that could maintain the initiated ice loss?

We will expand this point.

Lines 84-88: Some studies suggest that the ASL deepening is also driven by anthropogenic forcing, in particular the stratospheric ozone depletion (e.g., England et al., 2016) – albeit it is seasonally dependent. I think it would be worth mentioning.

We will add this point (see also below).

Lines 256-257: It is worth mentioning that the impact of stratospheric ozone depletion is mainly visible during austral summer, which could explain why the response to ozone is weaker than the response to GHGs (in addition to the fact that over the analyzed period, some decades are not impacted by ozone as mentioned by the authors).

We will add this point (see also response to reviewer comments).

Lines 260-261: Have you looked at the variance in the reconstruction and model simulations before computing the difference between the two for inferring the internally-generated variability? My feeling is that if the reconstructed variance is different from model simulations, the inferred internally-generated variance could be wrongly estimated (over and underestimated). This could directly impact the contribution of the internal and forced variabilities on the total change.

We agree that is crucial that the variability in the CESM1 simulations be consistent with that in the reconstruction if we are to derive the internally-generated trends using our technique. The pertinent question in the lines referred to here is whether the reconstructed centennial trends sit comfortably within the CESM1 ensemble of historical centennial trends. (The CESM1 ensemble spread in trends is caused by the modelled internal variability.) We show in Figure 3 that this is the case for LENS (see paragraph starting on line 337), and we also find this to be true for PACE (not shown, line 438). In particular, Figure 3 shows that the 'internal part' of the trends in the reconstruction matches that realised by the model. This model--reconstruction consistency is aided by the fact that we use CESM PACE simulations as the 'prior' in the reconstruction, but it is still a very positive result. We will add a note to line 260 referring the reader to this consistency test later in the paper.

Lines 265-266: I don't fully agree with the authors on the fact that the ASL deepening is internally generated. Figure 1b indicates an intensification of SAM (decreasing sea-level pressure around the Antarctic continent), driven by the forced variability. Yet, SAM strongly modulates the ASL. Therefore, to me, the ASL deepening is also driven by external forcings and not only by internal variability. In contrast with the forced response, the internal response is less spatially homogeneous. Figure 1c clearly shows a major role of tropical variability with the propagation of Rossby waves. Would it be possible to quantify the contribution of both the external and internal variabilities? To come back to

my previous comment, I think it is important to pay attention to the variance of the reconstruction and climate model simulations when assessing those contributions.

This important point relates to the definition of the ASL. If the ASL is defined in terms of its absolute central pressure, then it is certainly true that external forcing plays an important role in ASL trends, since external forcing drives a strong deepening of pressures over Antarctica (Figure 1b). However, if the ASL is defined in terms of local pressure anomalies, e.g. as an anomaly relative to the zonal mean pressure, then the influence of external forcing on ASL trends is much weaker, since external forcing drives trends that are primarily annular over Antarctica (Figure 1b). We prefer this 'local' definition of the ASL, since it relates more closely to the winds of interest, and separates the behaviour of the ASL from wider hemispheric signals such as the SAM. However this is a very important distinction and so the paragraph will be re-written to make this clear.

Lines 453-454: As mentioned in your previous paragraph, you should specify that the tropical Pacific cannot explain the entire variability of winds.

The tropical Pacific variability cannot explain the wind variability on centennial timescales, as detailed in the previous paragraph, and this paragraph is intended to consider interannual and interdecadal timescales. We will clarify this point.

Lines 616-617: As already mentioned above, I don't fully agree with the unique contribution of the internal variability to the ASL deepening. In the same paragraph, I think it is worth mentioning that the Tropical Pacific cannot explain all the internal variability (since it is in contradiction with the results of Holland et al. [2019]).

As detailed above, the role of internal variability depends upon the definition of the ASL, while the tropical Pacific cannot explain variability primarily on centennial timescales. We will clarify this paragraph.

---

## Author Comment (AC3)

This study investigates potential drivers of trends in near-surface pressure and winds over the Amundsen Sea, with a view to understanding the decline of the West Antarctic Ice Sheet over the past century. It uses a paleoclimate reconstruction of global fields alongside a series of large ensemble simulations with different forcings. The study concludes that internal climate variability has played a dominant role, particularly in the ice shelf and break region, with forced variability (greenhouse gases and ozone depletion) significantly contributing in the later 20th century and future.

This paper addresses an important issue in understanding and attributing the drivers of West Antarctic climate change. I enjoyed reading the study and found the text and figures to be clear and logically-structured. I think that this study is very suitable for publication in The Cryosphere and have just a few minor comments, which I hope the authors will find useful.

Minor comments:

1. In my view, the main caveat with this study is the reliability of the paleoclimate reconstruction used. In the short period of overlap between the satellite era (using ERA5) and the reconstruction, correlations of zonal winds are relatively modest at 0.35-0.6 (Fig 4). The paper does a good job of acknowledging and discussing this caveat, however I think that it would benefit from a little more detail on the reliability of the reconstruction. I would suggest that the authors include a comparison with the full ERA5 record (1950-present) in Fig 4. Although the reanalysis will also be substantially less reliable before the satellite era, there are in situ observations that will lend some skill during this time period, such that I believe the comparison is worthwhile.

As detailed in the response to reviewer 1, the reconstruction was extensively validated in the original paper by O'Connor et al. (2021). As well as comparison to modern reanalyses since 1979, that paper documents comparison to the station-based Marshall (2003) SAM index since 1958, and to the longer reanalysis datasets and paleoclimate reconstructions since 1900. In response to reviewer 1, we will further add a comparison of the reconstruction winds to those derived from the Fogt et al. (2019) spatially-interpolated Antarctic station dataset since 1957.

The key feature of all the above comparisons is that the data used for comparison are consistent throughout the period considered. As the reviewer notes, unfortunately modern reanalysis datasets suffer a discontinuity at the onset of satellite infrared sounding in 1979 (Hines et al., 2000; Marshall, 2003; Bromwich and Fogt, 2004; Marshall et al., 2022). This issue is particularly problematic in the Amundsen Sea, since its remoteness from any station data means that reanalysis fields are very weakly constrained before 1979, and hence there is a substantial discontinuity at that time. This discontinuity prevents us from considering the reanalyses over the full period from 1950 to present.

If we were to consider the reanalysis for only the period 1957-1979, the station data constraining the reanalysis would be relatively consistent. However, we feel that the reconstruction skill during this period is already validated, in a better way, by the above-mentioned comparisons to centennial reanalyses, Marshall (2003) SAM index, and the new Fogt et al. (2019) station pressure reconstruction.

It is clear from the comments of both reviewers that the validation of the reconstructed winds is an important concern and needs to be better explained in the paper. We propose to add a new paragraph of text to section 2.1 detailing the various validation tests carried out by O'Connor et al. (2021) and presenting our rationale as to why these analyses support the use of the reconstruction in our study. We also propose to add some new text to section 3.1.3 detailing the statistics of our comparison of the reconstruction to the Fogt et al. (2019) dataset for winds over the Amundsen Sea. This text will

respond to the reviewer's comment by adding further text to the paper detailing the following points: i) Modern reanalysis fields are discontinuous at 1979, particularly in this region; ii) The period 1957-1979 is validated using station data through their influence on longer reanalyses, the Marshall SAM index, and the new comparison to Fogt-reconstructed winds.

2. L411, Fig 6: The text discusses the correlations as statistically significant, indicating a strong relationship between the IPO and internal variability in zonal wind in the three regions. However, although the (annual) correlations are significant, I think it should also be mentioned that they are relatively small, meaning that the IPO can only explain at most ~25% of the variance in internal variability.

We will add this point.

3. L580: It is discussed here that future wind trends on the shelf are determined only by internal variability (not emissions scenario). It is also stated (e.g. L638) that mitigation of wind-driven ice loss will require strong emissions mitigation. This may perhaps be confusing, and so I would suggest some discussion here of whether winds over the deep ocean, shelf break, or shelf are expected to play the larger role in driving ice loss. If the shelf winds are thought to dominate, then these results might suggest emissions mitigation will have little impact on wind-driven ice loss.

We will add some new text to the paragraph starting on line 596 emphasising that the implications for ice-sheet melting are subject to two complicating factors. Firstly, the drivers of wind changes are seasonally varying, and so the attribution may be altered if ice-sheet melting is particularly sensitive to a particular season (in response to reviewer 1). Secondly, the contributors to wind changes are spatially varying, so the attribution is influenced by whether ice shelf melting is most sensitive to local winds over the Amundsen Sea, or remote winds over the deep ocean. Addressing these questions will be the subject of our future work.

**References**

Bromwich, D. H. and Fogt, R. L.: Strong trends in the skill of the ERA-40 and NCEP-NCAR reanalyses in the high and midlatitudes of the southern hemisphere, 1958-2001, Journal of Climate, 17, 4603-4619, 2004.

Fogt, R. L., Schneider, D. P., Goergens, C. A., Jones, J. M., Clark, L. N., and Garberoglio, M. J.: Seasonal Antarctic pressure variability during the twentieth century from spatially complete reconstructions and CAM5 simulations, Climate Dynamics, 53, 1435–1452, 10.1007/s00382-019-04674-8, 2019.

Hines, K. M., Bromwich, D. H., and Marshall, G. J.: Artificial Surface Pressure Trends in the NCEP–NCAR Reanalysis over the Southern Ocean and Antarctica, Journal of Climate, 13, 3940-3952, 2000.

Marshall, G. J.: Trends in the Southern Annular Mode from observations and reanalyses, Journal of Climate, 16, 4134-4143, 2003.

Marshall, G. J., Fogt, R. L., Turner, J., and Clem, K. R.: Can current reanalyses accurately portray changes in Southern Annular Mode structure prior to 1979?, Climate Dynamics, 10.1007/s00382-022-06292-3, 2022.

O'Connor, G. K., Steig, E. J., and Hakim, G. J.: Strengthening Southern Hemisphere westerlies and Amundsen Sea Low deepening over the 20th century revealed by proxy-data assimilation, Geophysical Research Letters, 48, e2021GL095999, 10.1029/2021GL095999, 2021.

---

## Author Response (AR1)

Overview – this study investigates historical (1920-2000) and future (2000-2080) changes in near surface pressure and winds over the South Pacific using a combination of a spatially complete paleoclimate reconstruction and a large suite of climate model simulations. It provides convincing evidence of the relative roles of internal variability and forced variability (from greenhouse gases and ozone depletion). Importantly, the paper also provides a much-needed possible narrative for the roles of natural variability followed by external forcing to understand historical variations in ice loss from the WAIS since 1940s.

The paper is very well written, easy to follow, and the scientific analysis in my perspective is sound. The team is to be commended on this excellent study, which is well conceived and an important scientific advancement. The authors note many caveats to the present study, which I also appreciate.

My main concern is that the paper is primarily based on gridded model or paleoclimate-based reconstruction data, and does not incorporate the wealth of observations other than ERA5 after 1979. It would be helpful to see the agreement between the reconstruction at least and pressure observations (available across all SH midlatitudes since 1920, and Orcadas since 1903) to see at least the agreement in South America / South Atlantic. For more complete investigation on the agreement (and for some aspects of the deepening of the ASL), comparisons could be made with Antarctic data after 1957 when trends seem quite large (making note to include the critical point of Byrd station in West Antarctica as another potential estimate of observed change near the Amundsen Sea apart from measurements along the Antarctic Peninsula). I do feel the point observations comparisons with the reconstructions would help to understand changes in observations apart from paleoclimate data and climate model data, and would round out the paper well (1 more figure), and provide further validation for the reconstruction that is not provided in the preceeding O'Connor et al. paper.

O'Connor et al. (2021) validate their reconstruction against other paleoclimate reconstructions of the 20th century, modern reanalyses datasets (since 1979), and longer-term reanalyses (since 1900). The longer-term reanalyses are constrained primarily by the station data mentioned above. O'Connor et al. (2021) also show that their reconstructed SAM index compares favourably to that of Marshall (2003), which is based solely on station pressure observations from Antarctica and the sub-Antarctic regions since 1958. Thus the reconstruction has been validated to some extent using the earlier observations referred to by the reviewer. In any case, we see no immediate reason why the reconstruction's skill in fitting modern reanalysis data would not apply to earlier periods, since the proxy data underlying the reconstruction are uniformly available throughout the 20th century.

We agree that directly comparing the O'Connor et al. (2021) reconstruction to each of the station pressure records around Antarctica would add detailed insight into the quality of the reconstruction at those locations. However, the applicability of such a comparison to the present study is not clear because there are no long-term station data anywhere near the Amundsen Sea. It is not clear how far any direct misfit to remote stations such as Orcadas or South America should limit our confidence in reconstructed Amundsen Sea winds. As a result we feel a systematic comparison at station locations is beyond the scope of the present paper.

To illustrate the issue of applicability of station data, we compared the reconstructed geostrophic winds used in the paper to geostrophic winds calculated from the surface pressure dataset of Fogt et

al. (2019), which is constructed by interpolating Antarctic station pressure observations. Figure R1 shows an extended version of Figure 4 from the paper, to which these winds derived from Fogt et al. (2019) have been added. The Fogt-derived winds are only positively correlated to ERA5 over the deep ocean, and do not provide a constraint over the shelf break or shelf. (We speculate that negative correlations over the shelf are reflecting the Amundsen Sea Low, whose pattern may not be reflected by the spatial kriging in the Fogt et al. dataset.) Over the deep ocean since 1957, the O'Connor and Fogt reconstructed winds are correlated at a similar level to the fit between the O'Connor and Marshall SAM indices (O'Connor et al., 2021). Prior to 1957 the fit between O'Connor and Fogt timeseries is much worse, which is unsurprising because the few direct station observations available during that period are very remote from the Amundsen Sea. We conclude that station data can usefully constrain Amundsen Sea winds, but only over the deep ocean since 1957.

It is clear from the comments of both reviewers that the validation of the reconstructed winds is a main concern and needs to be better explained in the paper. We have responded to these concerns by adding a new paragraph of text to section 2.1 detailing the various validation tests carried out by

---

## Author Response (AR2)

**Reviewer: A few points to note (please address in the paper on final revision):**

**1 - The Fogt et al. (2019) dataset is a krigging interpolation of station-based, seasonally resolved Antarctic pressure reconstructions. As such, the amount of data used in the krigging interpolation does not change in time, as suggested with more data entering in after 1957 on lines 381-382. Please fix.**

**2 - A limitation in the comparisons with Fogt et al. (2019) data is that you are averaging these seasonal data to compare with an annual mean from O'Connor et al. (2021). Further, it isn't clear if the Fogt et al. (2019) winds were corrected to maximize their correlation with ERA-5, like done with the O'Connor et al. (2021) winds.**

Response to both points:  The relevant text (from line 380) has been changed to

This validation may be extended further back in time using zonal winds derived from the interpolated station-based SLP reconstructions of Fogt et al. (2019). These seasonal reconstructions are averaged to produce annual-mean fields, and geostrophic winds are converted to near-surface winds using the simple correction described in section 2.1.